



# Calibration of a parsimonious distributed ecohydrological daily model in a data scarce basin using exclusively the spatio-temporal variation of NDVI

Guiomar Ruiz-Pérez[1], Julian Koch[2,3], Salvatore Manfreda[4], Kelly Caylor[5] and Félix Francés[6]

[1]Department of Crop Production Ecology, Swedish University of Agricultural Sciences, Uppsala, Sweden
[2]Department of Hydrology, Geological Survey of Denmark and Greenland, Copenhagen, Denmark
[3]Department of Geosciences and Natural Resources Management, University of Copenhagen, Denmark
[4]Department of European and Mediterranean Cultures, University of Basilicata, Matera, Italy
[5]Bren School of Environmental Science and Management & Department of Geography, UC Santa Barbara, California, USA
[6]Research Group of Hydrological and Environmental Modelling (GIHMA), Research Institute of Water and Environmental Engineering, Universitat Politècnica de València, Spain

*Correspondence to*: Guiomar Ruiz-Pérez (Guiomar.ruiz.perez@slu.se)

**Abstract.** Ecohydrological models provide a tool to investigate the mutual relationships between vegetation and the hydrological cycle. Ecohydrological modelling studies in developing countries, such as sub-saharan Africa often face the problem of extensive parametrical requirements and limited available data. Satellite remote sensing data may be able to fill this gap, but require novel methodologies to exploit its spatio-temporal information that could potentially be incorporated in ecohydrological model calibration and validation.

The present study aims to implement a distributed ecohydrological daily model in a data scarce environment with the support of remote sensing data. An automatic calibration procedure, based on Empirical Orthogonal Functions techniques, is proposed and applied in the Upper Ewaso river basin in Kenya. The model is calibrated only using NDVI (Normalized Difference Vegetation Index) data derived from MODIS. The obtained results demonstrate that: (1) satellite data of vegetation dynamics contains an extraordinary amount of information that can be used to implement ecohydrological models in scarce data dry regions; (2) the model calibrated only using satellite data is able to reproduce both the spatio-temporal vegetation dynamics and the observed discharge at the outlet point; and (3) the proposed semi-automatic calibration methodology works satisfactorily and it allows to incorporate spatio-temporal data in the model parametrization.

## 1 Introduction

Drylands cover 30% of the Earth's land surface and 50% of Africa (Franz *et al.,* 2010). Projections of the IPCC (Intergovernmental Panel on Climate Change, 2007) indicate that the extent of these regions will likely increase in the coming decades. Dryland expansion would have a considerable additional impact on water resources, which should be taken into account by water management plans (Franz *et al.,* 2010).



In water-controlled ecosystems, the vegetation assumes a critical role influencing all components of the hydrological cycle (Rodriguez-Iturbe *et al.,* 2001; Manfreda and Caylor, 2013). For instance, actual evapotranspiration (aET) may account for more than 90% of the annual precipitation in water-controlled areas (Zhang *et al*., 2016; Jasechko *et al.,* 2013). Montaldo *et al.* (2005) affirmed that the use of constant LAI (Leaf Area Index) values, commonly used in hydrological applications,

produces large errors in land surface flux predictions. Therefore, reliable estimates of spatio-temporal variations of vegetation patterns are vital to obtain trustworthy predictions of available water resources, given the strong control exerted on aET by the vegetation (Andersen, 2008). In this sense, ecohydrological modeling becomes essential in order to include the vegetation dynamics as an additional state variable (Rodriguez-Iturbe *et al.,* 2001).

Particularly, evidence of aET being prevalent in hydrological records of streamflow and water-table depth, i.e. available water

resources, has been observed in many studies (e.g. Gribovszki *et al.,* 2008). Recently, Tsang *et al.* (2014) showed that adding a better evapotranspiration scheme in a widely used runoff model improves streamflow predictions. Conradt *et al.* (2013), who compared three different strategies for deriving sub-basin aET, affirmed that incorporating spatial variation of aET in a semi-distributed model increases its robustness. Contrarily, Stisen *et al.* (2011) and others stressed that those improvements are not necessarily seen in the outlet hydrograph. However, it could also be interpreted in the inverse sense; good performances in

terms of the outlet hydrograph do not necessarily mean more reliable estimates of aET.

Actually, the stream flow record is traditionally the only observation used for the calibration of hydrological models, because it represents an integrated catchment response, and hence provides some inherent insight into the lumped behavior of the catchment (Stisen *et al.,* 2011; Koch *et al.*, 2016a; Michaud and Sorooshian, 1994; Reed *et al.,* 2004 or Smith et al., 2013). Nevertheless, several studies demonstrate that distributed hydrological models, which accurately simulate discharges at the

basin outlet, produce poor results at interior points. In that sense, Conradt *et al.* (2013) provided several examples for larger simulation errors within the model domain and they mentioned, among others, the outcomes given by Feyen *et al.* (2008), Merz *et al.* (2009) and Smith *et al.* (2012). Wi *et al.* (2015) pointed out that caution is needed when using an outlet calibration approach for streamflow predictions under future climate conditions. At this point, the idea of using spatial state variables with which to implement the new era of distributed (temporal and spatially) models emerged in order to balance the conceptual

distributed nature of this kind of models (Stisen et *al.,* 2011).

Remote sensing data are well suited to this purpose because while traditional observations generally consist of point data with little spatial support, remote sensing retrievals offer the capacity to provide detailed spatial coverage and pattern information (Franssen *et al.,* 2008, McCabe *et al.,* 2008 and Stisen *et al.,* 2011). Additionally, satellite data has the great advantage to be available everywhere. In this sense, among the wide range of possibilities in data scarce areas, the use of remotely sensed data

represents an excellent source that provides information with a fairly good spatial/temporal resolution (Yang *et al.,* 2012). In modeling, remote sensing data has been basically utilized in three different ways: (1) as forcing data (Xiao *et al.,* 2004; Yuan *et al.,* 2010; Samaniego *et al.,* 2011; and Stisen *et al.,* 2011), (2) as priori information of a particular parameter (Winsemnius *et al.,* 2008; Stisen *et al.,* 2011) and (3) for model's calibration and validation (see next section for an in-depth discussion of this point).



Since satellite imagery includes not only temporal information but also spatial patterns, a proper spatial model evaluation is required. As mentioned by Koch *et al.* (2015), spatial model evaluation is an active field of research not only in Hydrology but also in other disciplines besides it, as for example, Atmospheric Sciences (Brown *et al.,* 2011; Gilleland *et al.,* 2010). However, up to now, there exists no formal guideline on how to assess the goodness of fit of the spatial explicit model

predictions and little information can be found about how to handle with spatio-temporal data. That's why some authors such as Conradt *et al.,* (2013), Graf *et al.,* (2014) and Koch *et al.,* (2015 and 2016b) focused their efforts in order to develop and test metrics to be employed when spatio-temporal data is involved. For example, Koch *et al.* (2015) compared Kappa statistics, Fuzzy theory, and EOF-analysis in an attempt towards a true spatial model evaluation of distributed models. But, besides these efforts, there are only a limited number of spatial validation studies that fully embrace the availability of satellite remote

sensing data by means of true spatial performance metrics (Koch *et al.,* 2016b).

In this research, we applied the Empirical Orthogonal Functions (EOF) analysis to identify predominant spatial or temporal patterns in observed data (Graf *et al.,* 2014) by means of decomposition. Consequently, the EOF analysis is a useful methodology to investigate the spatio-temporal patterns of fluxes and states in the soil-vegetation-atmosphere continuum (Fang *et al.,* 2015). In particular, as mentioned previously, Koch *et al.* (2015) carried out a validation of a distributed model using

satellite based land surface temperature data by means of an EOF analysis. With other statistical purposes, the EOF analysis was used by Graf *et al.* (2014), Kim and Barros (2002) and Liu (2003). A fine scaled study was carried out by Drewry and Albertson (2006) who used the EOF analysis to associate spatial pattern in the errors of a canopy-atmosphere model with errors in the parameters. But, to our knowledge, the EOF analysis has not been applied in model calibration yet. In this research, we incorporated the EOF analysis in the calibration of a distributed model and proposed an automatized calibration procedure.

Having identified the importance of aET in the water cycle of drylands and, the potential of satellite data that is still largely unexploited (i.e. taking advantage simultaneously of both: spatial and temporal information). For this reason, this research wants to 'properly' apply satellite data in an ecohydrological model's calibration and validation and to develop a mathematical methodology to incorporate this particular kind of data and its spatio-temporal nature in model's automatic calibration.

## 2 Satellite data and model calibration/validation

As said previously, the applicability of remote sensing to calibrate and/or validate a model still remains a challenging task that may help to exploit information on spatial patterns contained in satellite imagery. To provide in-depth vision of this issue, a bibliographic survey of the ISI Web of Knowledge Science Citation Index database was undertaken using the following words combinations in the topic search: (1) satellite calibration, (2) satellite implementation, (3) satellite ecohydrological modelling, and (4) remote sensing ecohydrology. This search looked for each term in the title, abstract and keywords list in the publication

database (i.e. articles, letters and book reviews) of ISI-rated journals and conference proceedings since 2006 (we analyzed the last decade). From the total number of publications obtained by this search, only those that incorporated satellite data to



specifically models calibration were selected. We must acknowledge that the adopted searching method may have some limitations but it was complete enough for our purposes.

Ruiz-Pérez *et al.* (2016) discussed the applicability of satellite data during the calibration process comparing the results obtained by a parsimonious model calibrated only using satellite data against the results obtained by a complex model calibrated using field measurements at pixel scale. Also, Quevedo and Francés (2008) and Pasquato *et al.* (2015) calibrated and validated a parsimonious ecohydrological model at pixel scale using satellite information.

At catchment scale, Immerzeel and Droogers (2008) used satellite-based evapotranspiration in combination with observed streamflow to calibrate the semi-distributed SWAT. Zhang *et al.* (2009) concluded that multi-objective calibration of SymHyd model with streamflow and satellite-based aET produced better daily and monthly runoff compared to calibration with streamflow alone. More recently, Rientjes *et al.* (2013) calibrated a semi-distributed hydrological model using streamflow data and satellite-based aET. Regarding to other satellite products, GRACE (the US-German satellite mission) data have been used to calibrate both global and regional-scale surface hydrology models, in combination with stream discharge data (e.g. Lo *et al.*, 2010). Zhang *et al.* (2011) calibrated the AWRA-L model with streamflow, NOAA-AVHRR LAI and TRMM-MI (Tropical Rainfall Measuring Mission- Microwave Imager) soil moisture using multiobjective criteria. Only in few studies, the calibration was carried out exclusively with remote sensing data. For instance, Gutmann *et al.* (2010) calibrated landscapes hydraulic properties in the Noah land surface model using only MODIS surface temperatures from 14 different sites and using observed flux data for model verification. Also, Velpuri *et al.* (2012) modelled Lake Turkana water level only using satellite information. All these studies reach the same conclusion: including remote sensing data into the model calibration/validation improves the overall performance.

In general, from the total of reviewed publications, calibration only using satellite data was performed in the 47% of cases while a combination of satellite data and field measurements (specially, streamflow at the outlet) was used in the remaining contributions. Similar results were obtained regarding to the validation: 35.3% of publications adopts only field measurements (specially, historical streamflow) employing satellite data exclusively for the model calibration, 47% using a combination of field measurements and satellite data, 11.8% using only satellite data and one publication without any specification. But, more interesting is how the different calibrations were carried out. In most of the cited examples, a sort of multi-objective calibration was used adopting only some points/pixels to calibrate the entire catchment. Those points were selected randomly or by considering the knowledge about each study site. In other cases, lumped or semi-distributed models were implemented instead of fully distributed ones, considering aggregated values of the satellite data. In other words, the spatial heterogeneity of the basin is neglected and the full potential of satellite imagery, namely the information on spatial patterns, is not fully exploited.

Therefore, a method able to make use of the potential of the spatio-temporal information contained in remote sensed data is highly desirable as well as a calibration scheme which relies solely on remote sensing data will be greatly beneficial in modelling at data scarce catchments (Kunnath-Poovakka *et al.,* 2016). And these are the main objectives of this paper.



## 3 Study area and data

The Upper Ewaso Ngiro Basin is located in the Laikipia region of Kenya (Figure1). The basin is part of the Laikipia Plateau which lies between Mount Kenya (South East) and the Aberdare Mountains (South West). The basin has a drainage area of 15,200 km2, with the largest river being the Ewaso Ngiro. This region is characterized by distinct rainy and dry seasons. The

first rainy season occurs from March to May, while the second rainy season occurs from October to December. Both air temperature and precipitation patterns are heavily influenced by elevation. A full description of the precipitation patterns in the region can be found in Franz (2007).

Soil texture ranges from sandy clay to clay soils (according to the 1980 UNESCO Soil Map). Although the most characteristic landscape is savanna, higher elevations are dominated by forests and a large piece of land has been converted to cropland

(Franz, 2007). The remainder of the study region is classified as grassland, shrubland, and wooded grassland (savanna ecosystems).

For the modelling application, we used weather stations of the Natural Resource Monitoring, Modeling and Management Project (NRM3) of Nanyuki, Kenya illustrated in Figure 1. Daily precipitation and temperature from 1959 to 2003 were validated by Franz *et al.* (2010). Considering the available hydrological information, we selected a sub-basin with an area of

about 4,600 km2 for the present study (Figure 1). The selected catchment is equipped with a streamflow gauge at the outlet operating from 1980 to 2002.

The reference evapotranspiration (ET0) was calculated using the Penman Monteith equation with the simplifications proposed by Allen *et al.* (2006). This approach is extremely useful to describe the spatial distribution of solar radiation and to derive the ET0 maps during any phase of the year (Manfreda et al., 2013).

Regarding the satellite data, we incorporated the Normalized Difference Vegetation Index (NDVI) included in the MOD13Q1 and MYD13Q1 products provided by NASA (NASA Land Processes Distributed Active Archive Center (LP DAAC)). This satellite product is available from 2000 to present. For the coverage of the study site, the h21v08 and h21v09 tiles are required, where h and v denote the horizontal and vertical tile number, respectively. The MOD13Q1 and MYD13Q1 data are provided every 16 days at 250-meter spatial resolution. The used NDVI products (MOD13Q1 and MYD13Q1) are in level 3 that means

they don't contain raw satellite data. Actually, the NDVI indices are retrieved from daily, atmosphere-corrected, bidirectional surface reflectance. Specifically, these products use a MODIS-specific compositing method based on product quality assurance metrics to remove low quality pixels. From the remaining good quality NDVI values, a constrained view angle approach then selects a pixel to represent the compositing period (from the two highest NDVI values it selects the pixel that is closest-to-nadir). That's why assimilation approaches (such as Kalman filters) were not considered in this research.

At last, based on previous experience (Ruiz-Pérez *et al.,* 2016 and Pasquato *et al.* 2015) in a similar climatic conditions, we declined to use other products such as LAI or ET derived from MODIS because this kind of products are produced by models. And, for example, Ruiz-Pérez *et al.* (2016) found large discrepancies between the LAI provided by satellite and the LAI measured in field. At this point, we had no information to determine the accuracy of these particular models and the spatial





information used to implement them. In contrast, NDVI values are calculated by direct differences of spectrum bands, i.e. no models are involved and that's why we decided to use this latter product instead of satellite LAI and/or ET.

## 4 Model description:TETIS-VEG

The proposed model, called TETIS-VEG, is based on a distributed hydrological model called TETIS (Francés *et al.,* 2007) coupled with a dynamic vegetation model. Both models have simplicity in model structure in common. The used equations are as simple as possible in order to reduce the number of parameters and the number of parameters of each sub-model is specified in Table I. The sub-models are connected because the transpiration calculated in the hydrological sub-model depends on the leaf area index (LAI) simulated by the dynamic vegetation model. At the same time, the simulated LAI depends on the water stress which is calculated using the hydrological sub-model. A more detailed description of this link can be found in Pasquato *et al.* (2015) and Ruiz-Pérez *et al.* (2016). The hydrological sub-model can be used at different time scales (from minutal to daily timesteps) while the vegetation dynamic sub-model has to be applied at daily scale. Hence, the TETIS-VEG model must be used at daily scale.

### 4.1 The hydrological sub-model: TETIS

TETIS's conceptual scheme consists of a series of connected reservoirs or tanks, each one representing different water storages in the soil column: vegetation interception, first static soil layer (retained water by upper soil capillary forces, i.e., below field capacity plus water detention in surface puddles; evaporation and transpiration can occur), second static soil layer (retained water in deeper soil by capillary forces; only transpiration can occur), surface (for overland runoff), gravitational soil layer (upper soil water content above field capacity for interflow) and aquifer (for river baseflow). Vertical connections between reservoirs describe the precipitation, evapotranspiration, infiltration and percolation processes. The horizontal flows describe the three different hydrological responses that give the discharge at the catchment outlet: overland runoff, interflow and baseflow. A more detailed description of the TETIS model can be found in Francés *et al.* (2007) and GIMHA (2014).

The TETIS model uses a split-structure for the effective parameter value at each cell (Francés and Benito, 1995; Francés *et al.,* 2007). The effective parameter is calculated using a correction factor multiplied by the estimated value of the parameter in each cell using all the available information (land cover map, soil type map, DEM, depth of roots and soil layer, etc.) and expert's knowledge. Hence, we can distinguish between two parts: (1) the common correction factor for each type of parameter that takes into account the model and input errors and the temporal and spatial scale effects; and (2) the estimated parameter value at each cell. With the split-parameter structure, only nine correction factors are calibrated. Each one related to one of these estimated parameter maps: maximum static storage, reference evapotranspiration, infiltration capacity, hillslope velocity, percolation capacity, horizontal saturated conductivity for interflow, horizontal saturated conductivity for aquifer and river channel velocity.





## 4.2 The dynamic vegetation sub-model: LUE-model

The proposed dynamic vegetation sub-model is based on the concept of Light Use Efficiency (LUE) (Medlyn, 1998). The LUE is based on the proportionality between plant biomass production by terrestrial vegetation and absorbed photosyntetically active radiation (APAR) in optimal conditions. However, the LUE can be strongly affected by stress conditions. The key

factors contributing to the variation of this efficiency are: soil moisture content, air temperature (Landsberg and Waring, 1997; Sims *et al.,* 2006), and nutrient levels (Gamon *et al.,* 1997; Ollinger *et al.,* 2008). Since this model is designed to be used in water-controlled areas, the nutrient levels are not considered.

In the LUE-model, the water stress factor depends on the amount of water contained in the two static reservoirs and it is calculated according to Porporato *et al.* (2001). Basically, the stress factor is equal to 1 (maximum stress) if the water storage

is less than the water storage at wilting point; it is equal to 0 (minimum stress) if the water storage is higher than the water storage at critical point (plants start the stomatal closure); and, it varies from 0 to 1 using a potential function which depends on the wilting point, the critical point and an exponent set equal 2. Then, this stress multiplies the LUE index, reducing the efficiency when its value is lower than 1 (non-optimal conditions).

The LAI is simulated through the product between the leaf biomass, the specific leaf area (SLA) and the vegetation fractional

cover. Later, the LAI is used to calculate the transpiration in the hydrological sub-model according to the Eq. (1).

$$T_i = (ET_0 - EI) * \min(1, LAI) * \zeta * Z_i * fc \tag{1}$$

where Ti is the transpiration from the i soil layer, ET0 is the reference evapotranspiration, EI is the evaporation of the intercepted water, LAI is simulated by the model, Zi is the percentage of roots in the i soil layer and fc is the coverage factor. Therefore, the LUE-model has eight parameters to be calibrated: (1) Specific leaf storage (the maximum interception storage

is calculated as the product between the specific leaf storage and the LAI simulated by the model), (2) the LUE index (explained above), (3) the coverage factor, (4) the distribution of roots between the first and the second static storage layers, (5) the maximum LAI sustainable by the system (the simulated LAI is limited by a maximum), (6) the light extinction coefficient,k (this parameter is used to calculate the fPAR according to the Eq. (2), (7) the SLA and, (8) the optimal temperature (the stress factor also depends on the temperature).

$$fPAR = 0.95 * (1 - e^{-k*LAI}) \tag{2}$$

A complete description of the LUE-Model can be found in Pasquato *et al.* (2015).

## 5 Methodology

One of the main objectives of this research was to explore the potential of the satellite remotely sensed data for model calibration. Hence, the TETIS-VEG model was calibrated purely against MODIS NDVI. Therefore, modelling elaborations



were carried out into three different steps: (1) a manual calibration in order to obtain a first approximation of model parameters, (2) an automatic calibration based on the combined use of EOFs and a genetic algorithm in order to refine model parametrization and (3) a model validation carried out with both remote sensed data and traditional data (such as streamflow measurements). Considering that hydrological observations (precipitation and temperature) were available from 1960 to 2003

while the MODIS NDVI was available from 2000 to present, we decided to use the year 2003 as the calibration period and the period from 2000 to 2002 for validation. In order to avoid the effect of the initial conditions, we used one year as warming up period (the year 2002 and 1999 for model calibration and validation respectively).

For these purposes, we adopted the NDVI as a descriptor of the state of the vegetation assuming that LAI and NDVI are intimately related. Studies on various vegetation types, e.g., agroecosystems (Cohen *et al.,* 2003), grasslands (Friedl *et al.,*

1994), shrublands (Law and Waring, 1994), conifer forests (Chen and Cihlar, 1996), and broadleaf forests (Frassnacht *et al.,* 1997) have led to the general conclusion that the spectral vegetation indices such as NDVI have considerable sensitivities to LAI. Hence the relationship between NDVI and LAI has been documented by several authors (e.g., Gigante *et al.,* 2009). The relationship between LAI and NDVI can be considered linear for low values, while it becomes nonlinear for the higher values of the NDVI due to the greenness saturation (e.g., Turner *et al.,* 1999). In this case study, the maximum LAI values are around

2.0 – 2.5, according to Franz (2007), that are lower than the greenness saturation threshold. Therefore, the relationship between the observed NDVI and the simulated LAI is expected to be linear.

### 5.1 Empirical Orthogonal Function method (EOF)

The EOF method is used to analyze the spatio-temporal variability of a single variable but, comparison between different variables can also be performed using coupled EOF techniques (Björnsson and Venegas, 1997). The method decomposes a

dataset in a time series and spatial patterns. The method allows also to estimate a measure of the "importance" of each spatial pattern. We refer to the spatial patterns as the EOFs (in literature, they are also called as principal components), and to the time variation as loadings (in literature, there are several terms: expansion coefficient time series, expansion coefficients, EOF time series, principal components time series, etc.).

The EOF method is essentially a linear algebra methodology based on matrix transformation. The first step is the conversion

of the spatio-temporal data to be analyzed into a matrix. Basically, we construct a matrix (F) in which each column is the temporal variation of the data in a particular cell while each row represents the cells values during a particular time step. Usually, the second step is to compute the anomalies of the analyzed data which was not needed in this study because we used normalized data (for reasons that will be explained below).

The next step of the applied EOF method consists on the calculation of the spatial F's covariance matrix (R) according to Eq.

(3). Then, the eigenvalue problem is solved Eq. (4).

$$R = F^T * F \tag{3}$$

$$R * C = C * \Lambda \tag{4}$$





$\Lambda$ is a diagonal matrix containing the eigenvalues $\lambda i$ of R. The ci column vectors of C are the eigenvectors of R corresponding to the i-respective eigenvalues. Each of these eigenvectors can be regarded as a map which denote the EOFs (or principal spatial patterns). In what follows, we always assume that the eigenvectors are ordered according to the value of the eigenvalues. Thus, EOF1 is the eigenvector associated with the biggest eigenvalue. The fraction of the total variance in R explained by

EOFi is found by dividing the $\lambda i$ by the sum of all the other eigenvalues. The time evolution of an $EOF_j$ ( $\vec{a_j}$ ) is calculated according to Eq. (5). The components of this time vectors are referred to as loadings in this paper.

$$\vec{a_j} = F * EOF_j \tag{5}$$

Using the spatial covariance calculated according to the Eq. (3), the EOF technique provides three different results: the main patterns or EOFs, their time evolution whose components are called loadings and the portion of spatial variance explained by

each EOF which is calculated dividing each $\lambda$ by the trace of $\Lambda$.

## 5.2 Manual calibration

The manual calibration was done with a dual purpose. First, we wanted to test the applicability of the proposed TETIS-VEG model in the study basin. Second, we wanted to obtain a first approximation for the parameters and, at the same time, constrain the automatic calibration. Basically, this manual calibration consisted on the usual ad hoc method (manual adjustment of

parameter values) considering the Pearson correlation coefficient between the simulated LAI and the observed NDVI in a total of 32 different points inside the basin. These points were selected within homogeneous areas defined according to the main spatial patterns of the observed NDVI (EOFs) and the available maps of land cover, soil texture, DEM, slope and soil depth. In this case, the EOF analysis was used to identify the main spatial patterns of the observed NDVI. Once the main spatial patterns were identified, we combined our own human perception with the confusion matrices between the main spatial patterns

and the spatial maps of model parameterization. Confusion matrices are widely applied for map comparison in distributed modelling comparing actual to predicted values for each specific category defined previously (García-Arias *et al.,* 2016; Bennett *et al.*, 2013; Van Vliet *et al.,* 2013 among many others). Generally, the rows in the matrix represent the values predicted by the model, whereas the columns represent the actual values. By its nature, the confusion matrix is an overall measure for similarity between two categorized maps. However, the comparison of numerical maps is feasible if they are categorized

previously. In this research, we compared categorized map (land cover map, soil type maps, etc.) and the main patterns obtained by using the EOF methodology. That's why the main pattern of the observed NDVI (which is a continuous variable) was discretized according to the number of river basin features (such as land cover map, soil type map, etc.) and based on the similitude between the corresponding histograms. Once the discretization was done, by a cell-by-cell comparison of the discretized NDVI main pattern maps obtained after the EOF analysis and the available spatial maps, the confusion matrices

were built.



These confusion matrices allowed the calculation of the weighted kappa (k) coefficient (Cohen, 1968). This coefficient, whose maximum value is 1, representing a perfect agreement, was employed to identify which spatial maps (land cover map, soil type map, DEM, etc) were linked with the main patterns of the observed NDVI. Then, they were used in order to select the most appropriate points for the manual calibration.

## 5.3 Automatic calibration

The most innovative aspect of this automatic calibration was the incorporation of the EOF analysis as an objective function. As proposed by Koch *et al.* (2015), we decided to build one integral matrix concatenating both the observed and predicted data: the matrix contained the normalized values of the NDVI provided by MODIS and the normalized values of the LAI simulated by the model. In this way, the upper part of this matrix contained the temporal variation of the normalized observed

NDVI in all cells as columns while the lower part contained the temporal variation of the normalized simulated LAI in all cells as columns. We decided to use the normalized values of the NDVI and LAI because, although they are correlated, they differ in range.

However, normalization implies that some spatial information is lost. In order to avoid these losses, we added two rows in the matrix F: the first containing the difference between the temporal mean of the observed NDVI at a particular cell and the

general mean using the complete NDVI dataset; and the second with the same content referred to the simulated LAI. In this way, we included the spatial gradient of the observed NDVI and the spatial gradient of the simulated LAI. These two rows represents two additional maps included in the evaluation of the model performance. If they were similar, it would mean that the spatial gradient remains and is properly reproduced.

The number of pixels was 1,034,706. For the calibration period (year 2003), there were 44 NDVI maps (one each 8 days more

or less). Hence, the built integral matrix's size was 90 rows (44 + 44 + 2 additional rows) X 1,034,706 columns. After the construction of this matrix, the EOF analysis was applied obtaining: the EOF maps for the matrix containing both NDVI and LAI, the portion of variance explained by each EOF map and the loadings of each EOF map. The combined EOF analysis yielded orthogonal EOF maps that explained the combined intervariability and intravariability of both data sets. For each time step, the loadings express how much the respective LAI and NDVI map contribute to the direction of the corresponding EOF.

Hence, if the observed NDVI and the simulated LAI were completely correlated, the temporal evolution of the EOF maps for both, NDVI and LAI, would be essentially equal.

Basically, model calibration was carried out forcing the loadings of simulated and observed data to be close. The used objective function was based on that idea and it also took into account the portion of variance explained by each EOF in order to consider that the variance contribution decreases consecutively for the EOFs. The adopted error measure is described in following

equation:

$$Error = \sum_{i=1}^{k} w_i * \sum_{i=1}^{t} |load\_sim_{i,j} - load\_obs_{i,j}| \tag{6}$$



where Error is the objective function to minimize, $w_i$ is the portion of variance explained by the $EOF_i$, $load\_sim_{i,j}$ is the loading of the $EOF_i$ at time step j for the simulated data (in this particular case, the normalized LAI) and $load\_obs_{i,j}$ is the loading of the $EOF_i$ at time step j for the observed data (in this particular case, the observed NDVI).

The calibration was performed using a genetic algorithm called Pyevolve. This algorithm needs a seed (initial values of the parameters) and a searching boundary of the parameters to be calibrated. We used the results obtained after the manual calibration explained above as seed and made sure that the searching boundaries were wide enough (Table1).

After the automatic calibration process, in order to explore the outcomes of the proposed procedure, we calculated both the temporal Pearson correlation coefficient between the NDVI provided by MODIS and the LAI simulated by the TETIS-VEG

model in each cell and the spatial Pearson correlation at each time step. For the spatial and temporal correlation coefficients, we used the original values of both datasets (NDVI and LAI), not the normalized values as used by the EOF analysis. It is important to mention that the Pearson correlation coefficient between two datasets X and Y is positive if X and Y tend to be simultaneously greater than, or simultaneously less than, their respective means. Hence, the mean should be representative. For this reason, in the case of the spatial correlation coefficient, we decided to distinguish between the main land covers whose

means can be significantly different: tree, shrubs and grass.

### 5.4 Validation

The period selected for the model validation was of three years from 2000 to 2002. As during the calibration period (year 2003), there were data of precipitation, temperature and, also, NDVI provided by MODIS. To validate the model, we used the same performances indexes applied during the automatic calibration process. Keeping the parameter values obtained by the

automatic calibration, we built the matrix concatenating the normalized value of the observed NDVI and the normalized value of the simulated LAI with two additional rows used to incorporate the spatial gradient of both datasets as explained above. We also plotted these two maps and compared them as we did during the model calibration. Using the EOF techniques, we obtained the coupled EOF maps and their associated loadings and portion of variance explained by them. As during the calibration, we compared the deviation of the loadings for each EOF map and we calculated the Error function defined in Eq. (6).

We calculated the temporal and the spatial Pearson correlation coefficient as we did during the calibration period.

In addition to this, we also explore the reliability of the calibrated model in reproducing streamflow. In fact, during the validation period, the observed discharge at the outlet point was available unlike during the calibration period. Such condition was defined on purpose in order to avoid the use of any information regarding streamflow data during the calibration phase. This validation allowed exploring the reliability of the hydrological sub-model in reproducing the streamflow. This was an

extremely challenging task considering that the entire modelling structure had been calibrated only using vegetation data from remote sensing along with physical information about the basin.

With this aim, we calculated the Nash and Sutcliffe efficiency index (NS-Nash and Sutcliffe, 1970) and the bias (or volume) error (E) value between the observed and simulated discharges at the basin outlet. We also decided to strengthen our discharge





analysis by using the concept of flow duration curves (FDCs). FDCs are simple and powerful tools, commonly used in hydrology to describe the runoff regime in a river basin that can be representative of the model ability in reproducing the different components of the streamflow (e.g., Manfreda *et al.,* 2005). In fact, FDCs represent the relationship between magnitude and frequency of streamflows, providing thus an important synthesis of the relevant hydrological processes

occurring at the basin scale (Pumo *et al*., 2013). Actually, the shape of a flow-duration curve in its upper and lower regions is particularly significant in evaluating the stream and basin characteristics (Coopersmith *et al.,* 2012). The shape of the curve in the high-flow region indicates the type of flood regime the basin is likely to have, whereas, the shape of the low-flow region characterizes the ability of the basin to sustain low flows during dry seasons (Cheng *et al.,* 2012). Hence, the flow duration curve represents the full spectrum of variability in terms of their magnitudes (Wagener *et al.,* 2013).

**6 Results**

**6.1 Manual calibration**

As explained before, the main objective of this a priori manual calibration was the identification of the most appropriate points where the model could be tested. To do that, we identified the spatial main patterns of the observed NDVI and, then, we compared the EOFs with the spatial features of the river basin (such as: land cover map, DEM, soil type map, etc).

Using our own perception, we identified a certain relationship between the $EOF_1$ (which explained the 61.5% of the observed NDVI's spatial variance) and the land-use map. This potential relationship was supported by the K coefficient (described in the methodology section) that assumed a value of 0.34. This is not really high value but it showed the existence of a relationship between the two maps. I.e., there is a connection between the $EOF_1$ and the land-use map. Regarding to the $EOF_2$ (which explained the 10.5% of the observed NDVI's spatial variance), no connections with the basin physical characteristics were

found. It might contain a mix of several drivers and, therefore, it can't be directly linked to a single one. Contrarily, the $EOF_3$ showed a good agreement with the soil texture map (the K coefficient was 0.32). Therefore, we can state that the observed patterns of NDVI are strongly influenced by the spatial distribution of land cover and soil texture. In the following, we combined these two maps, extracted all possible combinations and selected randomly two points of each of these combinations obtaining 32 points covering all the catchment area.

When the manual calibration was stopped, the Pearson correlation coefficient between the observed NDVI and the simulated LAI was positive in 25 points of the 32 considered points. Hence, there were only seven points with negative correlation coefficient. All of them had in common the fact that they were located near to de Mount Kenya or Aberdare mountains (Figure 2).

Finally, Table I shows the obtained set of parameters. This set was used as seed during the automatic calibration. It must be

underlined that all parameters had values consistent with the reviewed literature (references embedded in Table I).





## 6.2 Automatic calibration

The proposed automatic calibration is based on the assumption that the closer the loadings of the simulated values are to the loadings of the observed values, the higher the similarity is. Calibration was carried out using a Pyevolve genetic algorithm using the objective function given in Eq. (6).

Calibration produced a good agreement between the observed and simulated loadings of the $EOF_1$ (upper part of the Figure 3, first graphic) while small deviation between the observed and simulated loadings related to the $EOF_2$ and the $EOF_3$. The loadings of the remaining EOFs were completely scattered mainly due to their corresponding low contribution (low weight) in the objective function of the automatic calibration process (Eq. 6). In this context, it is useful to remark that the $EOF_1$ explained more than 60% of the dataset spatial variance while the $EOF_2$ and the $EOF_3$ explained around 10% each. The

remaining EOFs explained less than 3% each, but in any case they were considered during the calibration process (weighted by the portion of variance explained by each one).

On the other hand, as mentioned in the methodology section, we also used three additional metrics to evaluate the model performance: (1) the temporal Pearson correlation coefficient evaluated in each cell, (2) the spatial Pearson correlation distinguishing between trees, shrubs and grasses computed at any time and (3) comparison of the called spatial gradient maps.

First, the temporal Pearson correlation coefficient between the observed NDVI and the simulated LAI was higher than 0.4 (left panel of Figure 4) in most of the catchment. The weakest correlations were obtained in the two higher areas of the basin near to the Mount Kenya and Aberdare Mountains with zero to negative values.

The spatial Pearson correlation coefficients were calculated excluding the regions with negative temporal Pearson correlation coefficient. Although slightly worse than the results in terms of temporal correlation, the mean spatial correlations were higher

than 0.45 for all main land covers: trees (mean=0.58), shrubs (mean=0.49) and grasses (mean=0.55) (Figure 5, upper panel). The best scores were obtained in cells classified as trees. In fact, the median was almost 0.60 and the variance was not high (standard deviation= 0.16). Contrarily, the cells classified as grasses obtained the worst results with the lowest median and the highest variance (standard deviation= 0.18).

Figure 6 (upper panels) shows the comparison between the maps which represent, in each cell, the difference between the

temporal mean and the general mean of the observed NDVI and the simulated LAI respectively. No great differences were found by comparing both maps indicating the good spatial performance of the ecohydrological model, at least from the vegetation point of view.

## 6.3 Validation

Similarly to the calibration process, the $EOF_1$ explained in validation more than 60% of the spatial variance while the $EOF_2$

and the $EOF_3$ explained around 10%. The remaining EOF maps are not presented because any of them explained more than 3%. The simulated and observed loadings of the $EOF_1$ were almost equals while the obtained results in relation to the $EOF_2$ and the $EOF_3$ were slightly worse (lower part of the Figure 3). However, it is important to stress both showed the same clear





temporal dynamics. Anyway, the resulted Error for the validation period was 4.03, just slightly worse than the Error for the calibration period. It must be considered that the Error value was calculated considering all EOFs (Eq. 6).

The temporal Pearson correlation map between simulated LAI and NDVI showed the same pattern observed in the calibration period: the two areas located near to the Mount Kenya and the Aberdare Mountains had a temporal correlation coefficient equal to zero or negative. However, in more than 80% of the catchment, this coefficient was between 0.3 and 0.9 (right panel of Figure 4).

Regarding to the spatial Pearson correlation coefficient between simulated LAI and NDVI in the three main land cover, the results were not as good as the results obtained in terms of temporal correlation. Nevertheless, there were no negative spatial correlation coefficients at any time step. In the case of shrubs and grasses, the mean and median were almost 0.4 while the corresponding ones for the trees were around 0.35 (Figure 5, lower panel). The variance obtained during the validation period was narrower than the obtained during the calibration period for the three land covers: trees, shrubs and grasses. Furthermore, the spatial pattern of LAI was, as for the calibration period, well captured by the model (see the lower panels in Figure 6). The cells with high differences between their own temporal mean and the general mean were consistent in both maps.

Finally, since there was observed discharge at the basin outlet during the years 2000, 2001 and 2002, it was possible to compare the discharge simulated by the model against the observations. The volume error (E) was equal to -0.40 while the NS index was equal to 0.32. E is strongly affected by the results obtained at the beginning of the validation period, probably due to the absence of information regarding the initial conditions. Although we used a year as warming-up period, the simulations improved only from 2001. In fact, having calculated the performances indexes in each year, the E decreased from -0.88 in 2000 to only -0.17 during the year 2002 (Figure 7). Regarding to the NS index, the worst result was also obtained for the first year and it improved from a negative value in 2000 to 0.35 during the year 2002, as one should expect considering the visual comparison in Figure 7. This trend is emphasized in the plot of the FDCs (Figure 8) where the underestimation in the first two years is clearly highlighted. The first panel compares the FDC of observations and simulations within the whole period while the following three panels compared the corresponding FDCs within the 2000, 2001 and 2002. In these plots, the simulation seems to interpret closely hydrological response in the year 2002.

# 7 Discussion

From the a priori manual calibration step up to model validation, it was possible to identify a behavioral pattern which would be also observed during the following automatic calibration and validation steps: the $EOF_1$ explains more than 60% of the spatial variance, the $EOF_2$ around 10%, the $EOF_3$ around 5% while the remaining EOFs could be considered negligible. The fact that the $EOF_1$ and $EOF_3$ of the observed NDVI was related to the land cover and soil type maps respectively was consistent with what one can expect as long as the NDVI is an indicator of vegetation dynamics.

After the automatic calibration, the model fitted the loadings of the $EOF_1$ and its accuracy is slightly worse regarding the second and third EOFs. Thus $EOF_1$ captured the predominant pattern that was found in both, the observed NDVI and the



simulated LAI data. Furthermore, on one hand, the temporal variation of the $EOF_1$'s loadings seemed to be related to the two typical growing seasons in the catchment: the first one during March-May and the second one during October-December (Franz *et al.,* 2010) (Figure 3). On the other hand, the loadings of $EOF_2$ and $EOF_3$ were not strongly connected with any feature. The loadings of the remaining EOFs were scattered which implies that mainly measurement and model noise were covered by

these EOFs. Nevertheless, the accuracy between the observed and the simulated loadings could be considered satisfactory.

The weakness of the proposed calibration methodology is that, although the associated weights to the loading deviation in Eq. (6) are needed, they are also misleading some spatial information. New ways to weigh the loading deviations must emerge in future researches as proposed by Koch *et al.* (2015). In fact, due to the portion of variance explained by the $EOF_1$, this first main pattern controlled the calibration process. In future applications, the proposed error index could be improved if we didn't

want the $EOF_1$ to dominate the calibration process or we wanted to emphasize a particular EOF map. A popular method for deciding which EOF to keep and which to discard is to use 'selection rules'. Basically, there are three classes of selection rules depending on whether they focus on the amount of variance explained by each EOF, the loadings or the EOF maps (Preisendorfer, 1988). Other option could be to rotate the EOFs as proposed by Bonaccorso *et al.* (2003). Basically, as each rotated EOF will not explain the same variance of the unrotated one, this approach would be an option to use different

combinations of EOFs which explain different amount of variance in order to reduce the influence of the $EOF_1$. However, the real fact is that the variability captured in $EOF_1$ is predominant and explains more than 60% of the total variance and should thus be weighted more.

Actually, the automatic calibration process works satisfactorily as shown by the additional metrics: temporal Pearson correlation coefficient, spatial Pearson correlation coefficient in the main land covers and the comparison between the gradient

maps. In terms of spatial Pearson correlation coefficient, the weakest values were obtained in the higher portion of the basin near to the Mount Kenya and Aberdare Mountains, while the remaining cells within the study area showed a good agreement between observed NDVI and simulated LAI. This same behavior was also observed when calibrating manually.

Two reasons could explain such results. First, the observed NDVI in some cells of those areas had a really bad quality testified by the unrealistic oscillations of the NDVI from 0.8 to 0.1 (even zero) in just one week. These unrealistic oscillations could be

produced by the presence of clouds over the area near to the mountains. The second reason is related to the conceptual limitation of the proposed model. The TETIS-VEG was designed to be used only in water-controlled areas. Franz (2007) combined the fractional woody cover and the mean annual precipitation (MAP) in order to provide some insights as to the limiting resources in the basin. Two different behaviors could be observed indicating the point in which water had a smaller influence. The transition point occurred approximately around 800mm/year. Physically, the transition point is believed to be a good

approximation of the transition from a water-controlled ecosystem to a nutrient-controlled ecosystem. Franz (2007) affirmed that the high-latitudes (where Mount Kenya and Aberdare Mountains are included) were nitrogen limited ecosystems.

With the exemption of these two areas, it is clear that there exists a strong correlation between NDVI and LAI, i.e the model can capture the temporal dynamic of LAI but it does not necessarily mean the magnitude of LAI is reasonable. This last point




was proven by calculating the spatial Pearson correlation and the comparison between the gradient maps. No differences and good agreements were observed along the main land covers: trees, shrubs and grasses.

Finally, for the automatic calibration, there were four parameters which changed substantially (in relative terms) in comparison to the values obtained during the manual calibration: the correction factor of the maximum static storage, the correction factor

of the reference evapotranspiration, the factor related to the distribution of roots between the first and second static storage layers and the maximum LAI sustainable by the system (Table I). These parameters affect directly on the transpiration process and on the amount of available water to be consumed by the plants. In any case, all obtained values were consistent with the reviewed literature (embedded in Table I). All of them are completely included in the searching boundary used during the automatic calibration and there were not reasons to think we should use wider ranges.

Similar results were obtained regarding to the EOF analysis and the additional metrics computed within the validation period. In fact, the validation process confirmed: (1) the model was able to capture completely the $EOF_1$ while the model performance worsened in the following two EOFs, (2) the simulated LAI and the observed NDVI were temporally correlated in most of the catchment and (3) the spatial distribution of LAI was consistent as shown by the comparison between the gradient maps and the value of the spatial Pearson correlation coefficient at any time.

An additional interesting outcome provided by the validation was the comparison between simulated and observed hydrograph at the outlet point. Stream flow simulations presented were promising, but not completely convincing. It is obvious since the model parameters were calibrated on NDVI data, i.e. the model was calibrated on vegetation dynamics. That's why the direct comparison between hydrographs could be too exigent when considering nothing was known about the parameters involved in hydrological processes not linked with vegetation, as the river flow routing or aquifer discharge.

Therefore, we strengthened our discharge analysis by using the concept of FDCs. By graphical comparison (Figure 8), it could be observed that the model is able to reproduce the shape of the observed FDC, while some discrepancies were found in terms of magnitude. However, its performance improved considerably year over year. Since the FDC shape is an important synthesis of the relevant hydrological processes occurring at the basin scale, this result pointed out the capability of the proposed model calibration methodology to reproduce the main hydrological behavior of the study basin.

## 8 Conclusions

The main two objectives of this research were: (1) to explore if it is possible to calibrate and validate an ecohydrological model only using satellite information, and (2) to incorporate spatio-temporal data about a model state variable into an automatic calibration process. In order to tackle these questions, a parsimonious distributed ecohydrological model was calibrated using exclusively NDVI data provided by MODIS. A methodology based on the EOF analysis was proposed to carry out the model

manual and automatic calibration. Finally, the results were validated using satellite data referring to different periods and, also, the observed discharge at the basin outlet which was not used for calibration.




In general, the proposed model is able to reproduce properly the vegetation dynamics and the observed streamflow. Regarding to the first objective of this work, the results highlighted the enormous usefulness of satellite data. It was possible to completely implement the hydrological and the vegetation components of TETIS-VEG daily model only using NDVI data and the model validation can be considered satisfactory. This fact is a promising conclusion particularly for ungauged basins because it means

that satellite data could be used in order to obtain river discharges at certain conditions. At the same time, this result also shows the key role played by vegetation in water-controlled areas such as the upper Ewaso river basin in Kenya. Of course, the time step also was a relevant factor in the transfer of information from satellite NDVI to hydrological parameters: at daily time step the runoff propagation was not relevant in this case study and the model was able to reproduce the flow duration curve with no information about the parameters involved in the river flow routing process.

The proposed automatic calibration was completely designed in order to incorporate spatio-temporal data in order to take the maximum advantage of the available satellite data. After calibrating, the simulated vegetation patterns display good agreement with measured NDVI in most of the basin except for some portions at higher altitudes. This non-satisfactory result may be due to the bad quality of the NDVI data and/or the limitation of the vegetation sub-model (that was specifically designed for semiarid regions).

Nowadays, there is a grand availability of remote sensing information (not only satellite) concerning spatial state variables and more information will be available in the future. Many efforts are being done to improve the quality and quantity of remote sensing data (drones, better devices, etc.). And, the scientific community must also be ready to work with different kinds of information (temporal, spatial and spatio-temporal) simultaneously. If we want to be efficient, we have to identify the best way to use all of this new available information, not only for data assimilation, but also and more important from our point of

view, for model calibration and validation.

**Acknowledgements**

The research leading to these results has received funding from the Spanish Ministry of Economy and Competitiveness and FEDER funds, through the research projects ECOTETIS (CGL2011-28776-C02-014) and TETISMED (CGL2014-58127-C3-3-R). The collaboration between Universitat Politècnica de València, Università degli studi della Basilicata and Princeton

University was funded by the Spanish Ministry of Economy and Competitiveness through the EEBB-I-15-10262 fellowship. The MODIS data were obtained through the online Data Pool at the NASA Land Processes Distributed Active Archive Center (LP DAAC), USGS/Earth Resources Observation and Science (EROS) Center, Sioux Falls, South Dakota (https://lpdaac.usgs.gov/get_data).



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





**Table I. Summary of the initial values, the search range and the final value of the parameters or correction factors of both sub-models (hydrological and dynamic vegetation sub-models) as well as the units and the reviewed references.**

| Model | Correction factor or parameter* | Units | | Initial Value | Search Range | Final Value | References |
|---|---|---|---|---|---|---|---|
| HYDROLOGICAL SUB-MODEL | FC1-Maximum Static Storage | [-] | | 1.00 | [0.5,2.5] | 1.80 | [1] |
| | FC2-Evapotranspiration | [-] | | 0.70 | [0.7,1.2] | 1.05 | [1] |
| | FC3-Infiltration | [-] | | 0.20 | [0.01,2] | 0.12 | [1] |
| | FC4-Slope velocity | [-] | | 1.00 | [0.1,1.2] | 1.00 | [1] |
| | FC5-Percolation | [-] | | 0.08 | [0.001,2] | 0.05 | [1] |
| | FC6-Interflow | [-] | | 140.00 | [0.001,100000] | 150.12 | [1] |
| | FC7-Deep percolation | [-] | | 0.06 | [0.001,2] | 0.04 | [1] |
| | FC8-Connected aquifer | [-] | | 20.00 | [0.001,100000] | 16.82 | [1] |
| | FC9-Flow velocity | [-] | | 1.00 | [0.2,1.2] | 1.00 | [1] |
| VEGETATION SUB-MODEL | Specific Leaf Storages | mm | Tree | 0.50 | [0.5,3] | 0.43 | [2],[3],[4] |
| | | | Srhub | 2.00 | [0.5,3] | 2.00 | |
| | | | Grass | 2.00 | [0.5,3] | 2.00 | |
| | LUE | kg/m² MJ | Tree | 1.50 | [1.2,2.5] | 1.14 | [5],[6] |
| | | | Srhub | 1.50 | [1.2,2.5] | 1.14 | |
| | | | Grass | 1.50 | [1.2,2.5] | 1.71 | |
| | Coverage factor | [-] | (**) | 0.80 | [0.1,1.0] | 0.90 | [3],[4] |
| | Distribution of roots | [-] | Tree | 0.30 | [0.0,1.0] | 0.10 | [3],[4],[7] |
| | | | Srhub | 0.5 | [0.0,1.0] | 0.20 | |
| | | | Grass | 0.7 | [0.0,1.0] | 0.34 | |
| | Maximum LAI | m²/m² | Tree | 2.50 | [0.5,3.5] | 3.10 | [5],[8],[9],[10] |
| | | | Srhub | 2.00 | [0.5,3.5] | 2.00 | |
| | | | Grass | 1.00 | [0.5,3.5] | 1.50 | |





| Light extinction coefficient | [-] | All | 0.50 | [0.4,0.6] | 0.52 | [11] |
|---|---|---|---|---|---|---|
| SLA | m$^2$/kg | Tree | 4.00 | [2.0,5.0] | 4.00 | |
| | | Srhub | 6.00 | [4.0,20.0] | 10.00 | [5],[12] |
| | | Grass | 6.00 | [6.0,50.0] | 30.00 | |
| Optimal temperature | °C | All | 16 | [10,30] | 18 | [11] |

(*) Regarding to the hydrological sub-model, the table shows the value of the correction factors while regarding to the vegetation sub-model, the table shows the parameter values

(**) The coverage factor depends on the location. The value in the table is the mean value. We used the reported information by [3] and [4].

[1] GIMHA Team, 2014

5  [2] Van Dijk *et al.,* 2011

[3] Franz *et al.,* 2007

[4] Caylor *et al.,* 2006

[5] TRY Database (www.try-db.org)

[6] Yuan *et al.,* 2007

10 [7] Le Roux *et al.,* 1995

[8] Pasquato *et al.*, 2015

[9] Ceballos and Ruiz de la Torre, 1979

[10] López-Serrano *et al.,* 2000

[11] Ruiz-Pérez *et al.,* 2016

15 [12] Castro de Costa *et al.,* 2014





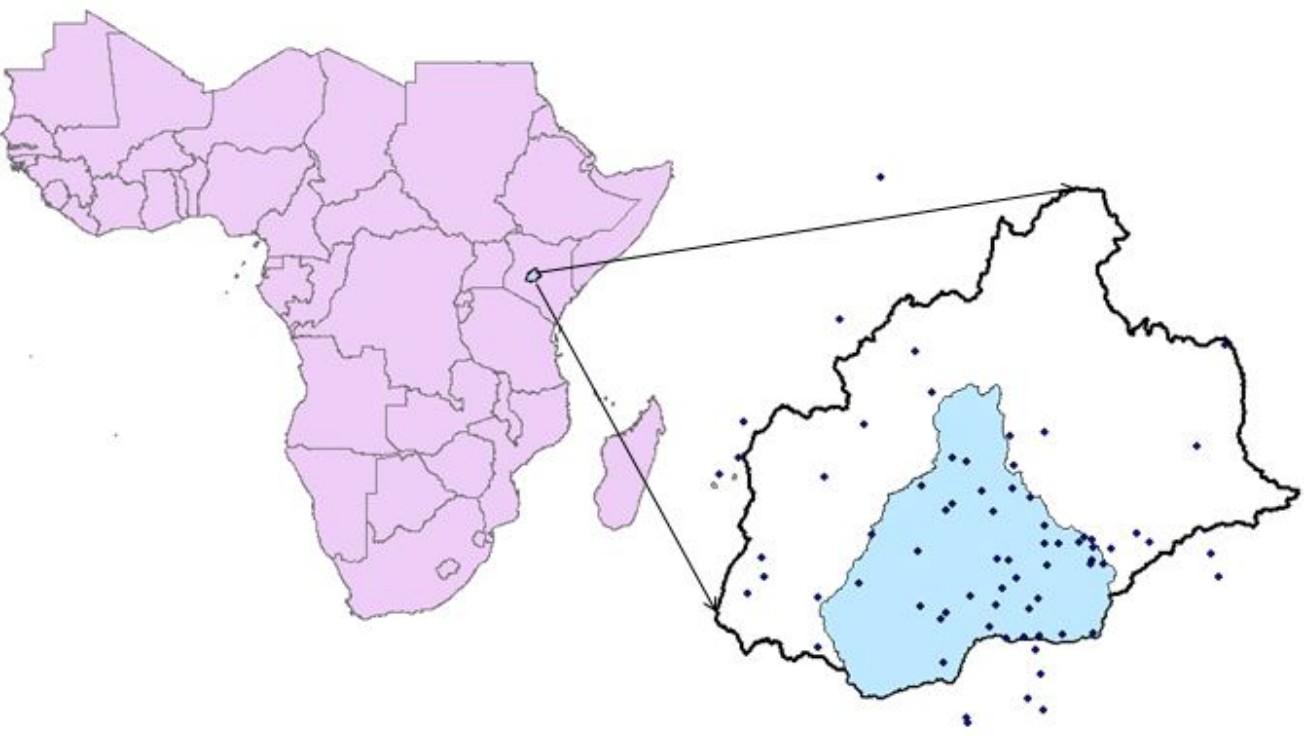

**Figure 1. General map location of the Ngiro river basin within the boundaries of the Sub-saharan Africa. The study sub-catchment (in blue) was selected because the density of the rainfall stations (points in dark blue).**





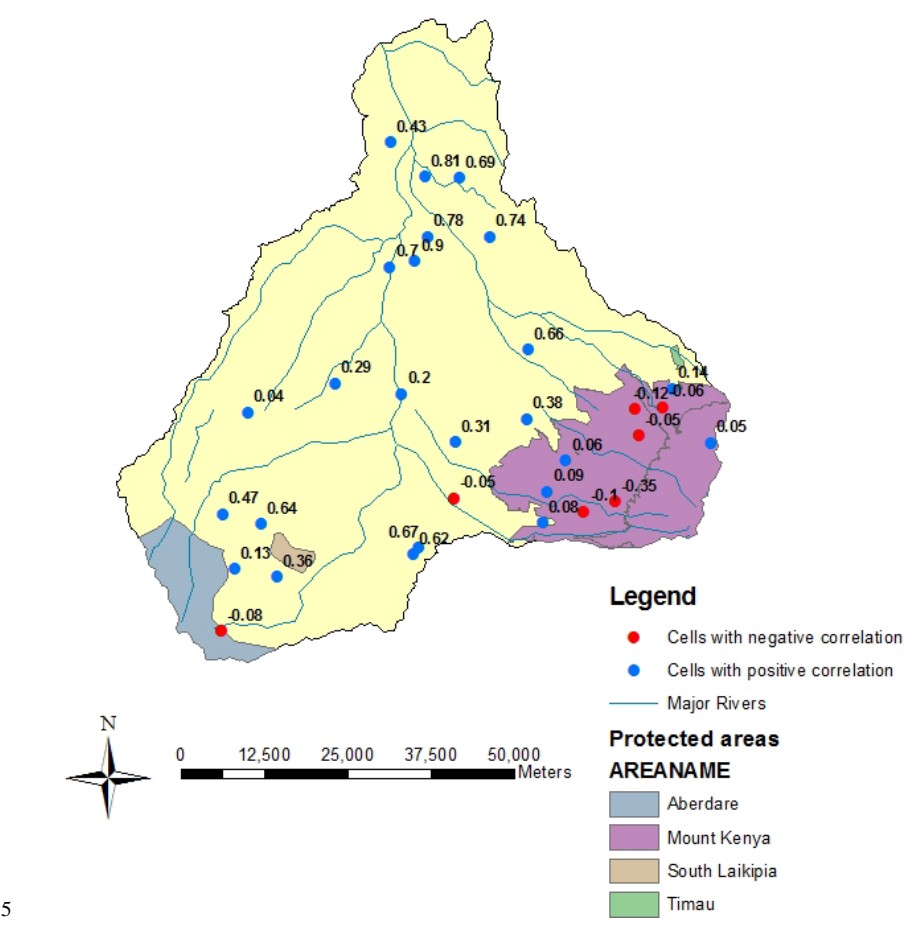

6   **Figure 2. Location of the points where the manual calibration was carried out. The value of the Pearson**
7   **correlation coefficient between the satellite NDVI and the simulated LAI appears together to the point used to**
8   **the manual calibration of the model**





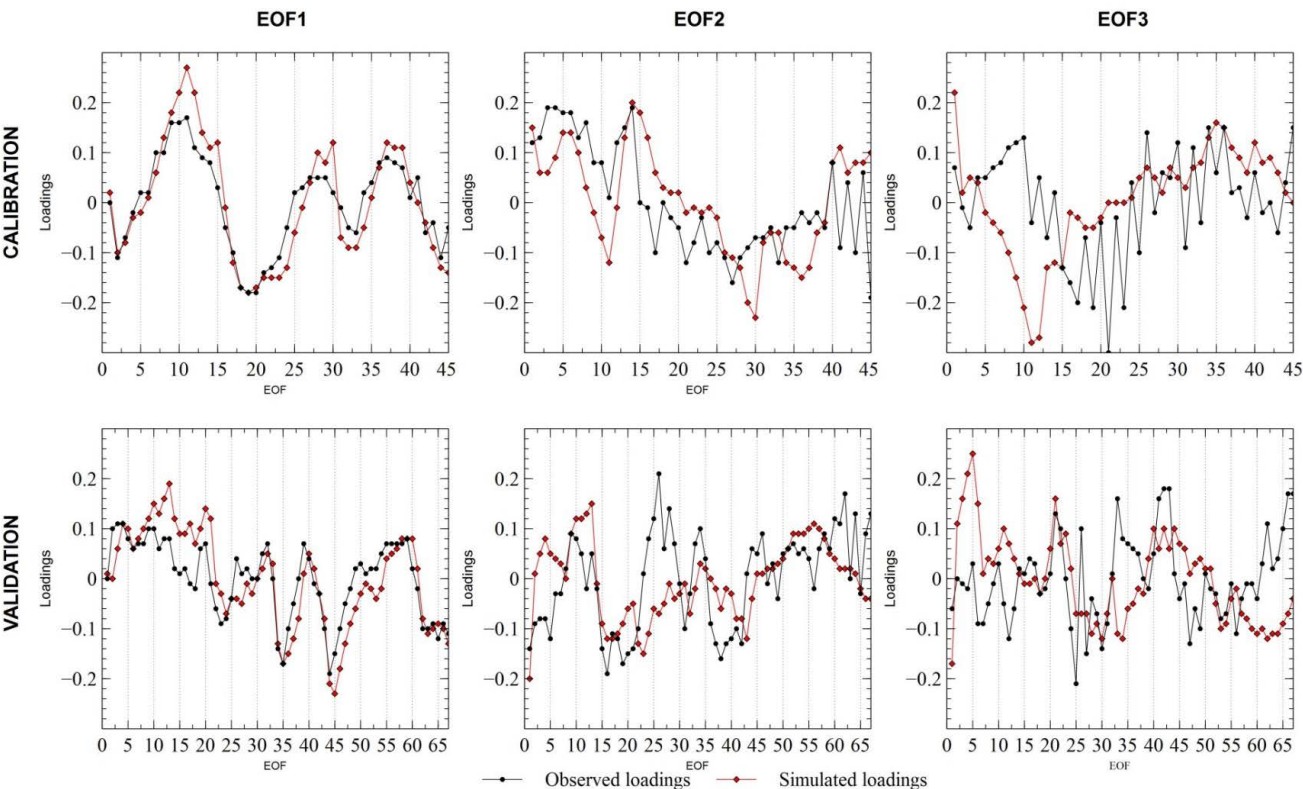

**Figure 3. The three first EOFs during the calibration (upper part) and during the validation (lower part) are represented. The y-axes reflect the unitless loadings of each EOF. The x-axes reflect the time step.**





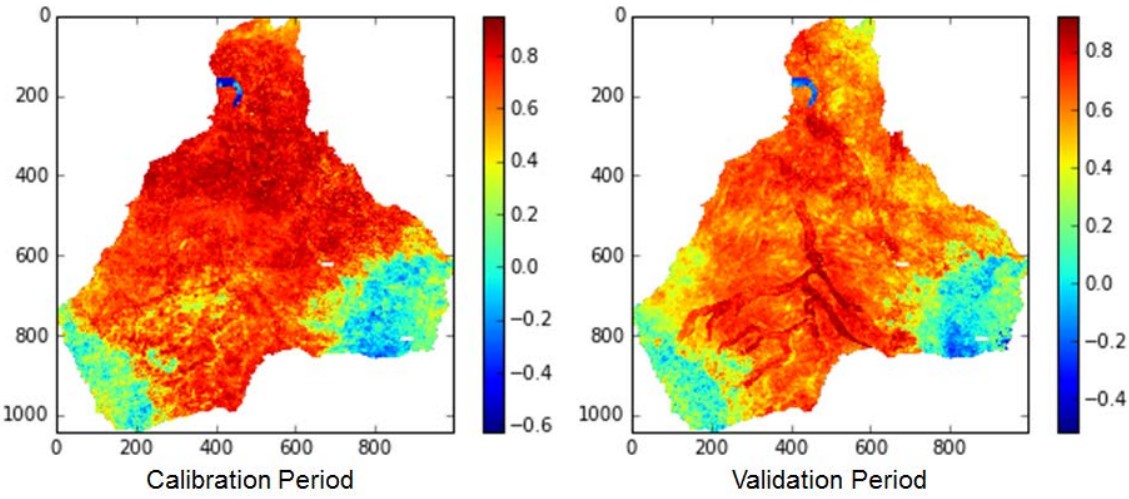

**Figure 4.** Temporal Pearson correlation coefficient between the NDVI provided by MODIS and the LAI simulated by the model during the calibration and validation period. The two areas with negative values correspond to the Mount Kenya and Aberdare Mountains.





**Figure 5. Spatial Pearson correlation coefficient during the calibration (upper panel) and during the validation (lower panel) distinguishing between the main land covers: tree, shrubs and grass. The whiskers were calculated according to the 98% percentile and the outliers were plotted as x. The median is the line inside boxplot and the mean is the quadrangle.**





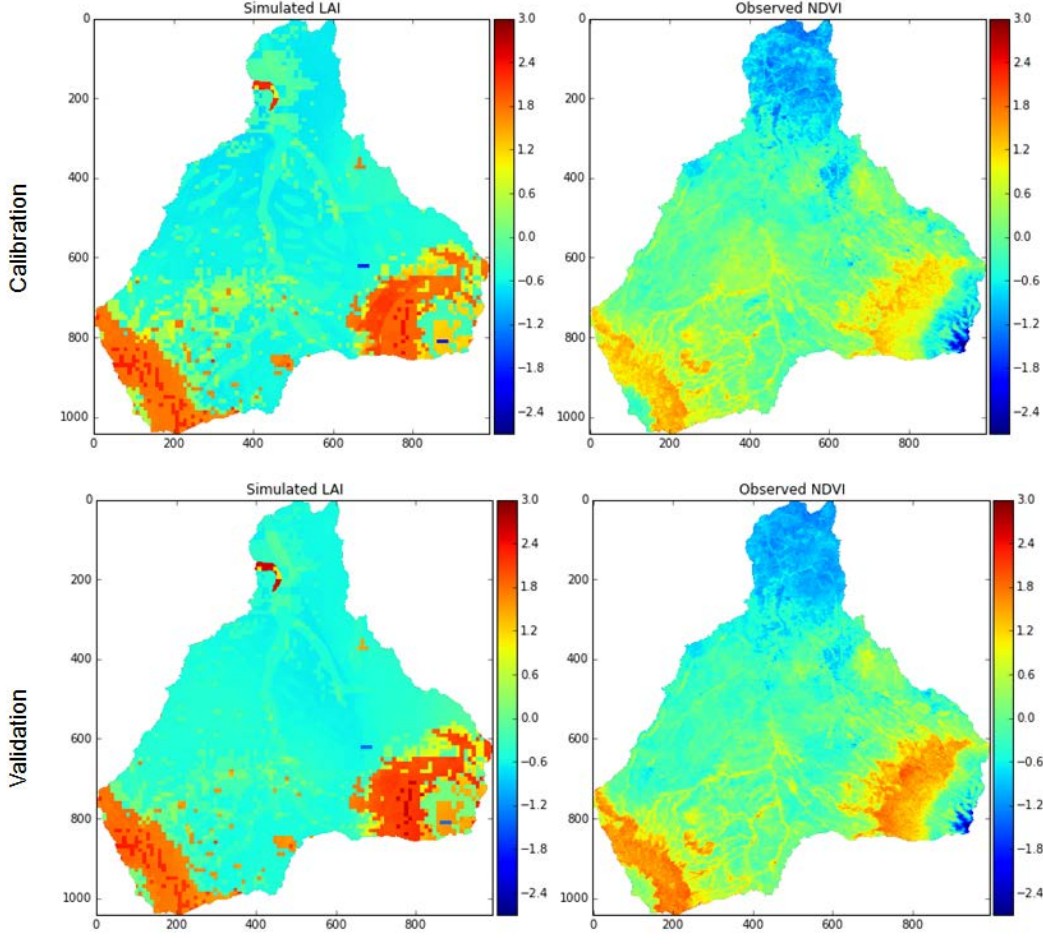

**Figure 6. Comparison between the maps where each pixel contains the difference between the temporal mean calculated in this particular pixel and the general mean calculated using the all dataset of the simulated LAI (left) and observed NDVI (right) in both periods: calibration (upper panels) and validation (lower panels). This difference is a measure of spatial gradient of both variables (LAI and NDVI).**





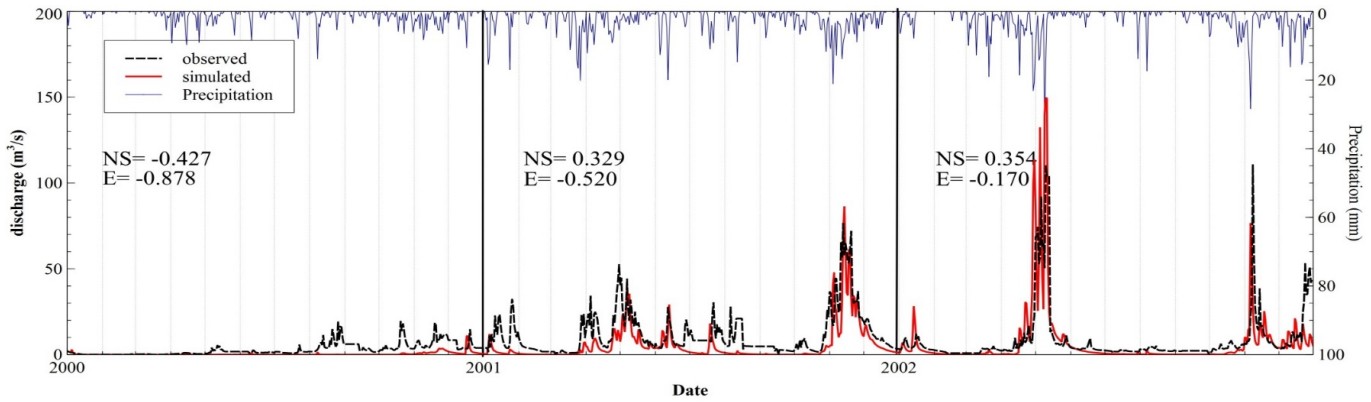

**Figure 7. Time series of rainfall and observed and simulated daily discharge (m³/s) during the validation period (2000,2001 and 2002)**



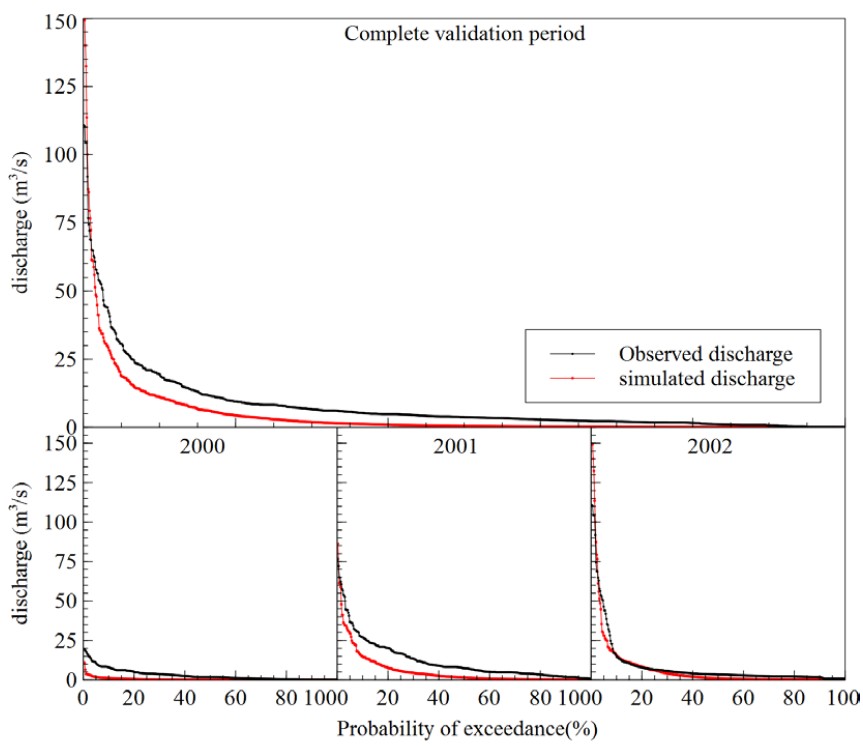

**Figure 8. Observed (in black) and simulated (in red) flow duration curves for the whole validation period (upper panel) and for the corresponding three years in isolation (lower three panels): 2000, 2001 and 2002 respectively**