# Peer review of "Calibration of a parsimonious distributed ecohydrological daily model in a data scarce basin using exclusively the spatio-temporal variation of NDVI"

_Hydrology and Earth System Sciences, 2016_

## Referee Comment (RC1) · S. Gharari (Referee) · 16 Jan 2017

I have read the manuscript with much interest. I found it very relevant. There are handful of models trying to mimic the vegetation dynamic as well as hydrological fluxes such as transpiration and streamflow with much more complexity. This study is relevant in the sense that it explores the possibility of using simpler models for mimicking vegetation dynamic.

To exploit the opportunity of the online discussion, I would like to ask the authors couple questions. Short answers to these questions will help me better judge the manuscript.

1- Are the model parameters different from cell to cell? If yes, which parameters are identical and which parameters are different?

2- I did not understand how the model calculate the LAI which then is used to calculate the transpiration?

3- Maybe I missed, but what is the resolution of the implemented model?

4- How did the manual calibration help to find the best parameters? How the parameters' ranges have been constrained? In table 1, LUE tree and shrub is out of specified range (Shrub is misspelled).

5- A clearer explanation regarding EOSi would be appreciated. What does different i exactly mean?

6- How would be the model performance with and without calibration on observed satellite data? Any gains or losses there? This would be great to be addressed.

With kind regards

Shervan Gharari

---

## Short Comment (SC1) · 17 Jan 2017

Thank you for your kind response and your stimulating questions. It is always a pleasure to hear that your work is causing interest and is providing new insights. Here, we have tried to respond as short and clear as possible each question formulated by you.

1- Are the model parameters different from cell to cell? If yes, which parameters are identical and which parameters are different?

To answer this question is important to better understand the concept of split-structure for the effective parameter value at each cell. This calibration strategy consists on an

application of a scalar multiplier to each prior parameter field (specified from data describing watershed characteristics: soils, vegetation, topography, land use, etc.) and to estimate a "best" value for this multiplier via calibration. This so-called "multiplier" approach makes the assumption that the prior parameter field properly describes the spatial pattern of parameter variation (the pattern of relative magnitudes from cell to cell), but that the magnitudes of all the parameter values must be adjusted to achieve a better simulation of the model response. Hence, the effective parameter at each cell (i.e. the parameter value used when running the model) is compounded by two parts: (1) a common correction factor for each type of parameter that takes into account the model, information and input errors and the temporal and spatial scale effects; and (2) the estimated parameter value at each cell. Hence, the parameters are different from cell to cell while the correction factor is common from map to map. The estimated parameter values were extracted from the field work done and presented in the doctoral thesis by Franz (2007) and following the recommendations provided by the TETIS model's support team. Two of the authors are actually active members of this team and we also used our own experience.

2- I did not understand how the model calculate the LAI which then is used to calculate the transpiration?

The LAI is calculated by the dynamic vegetation sub-model called LUE-Model. The LUE-Model computes the leaf biomass (Bl) according to the following equation:

$$(dB\_l)/dt = (LUE*\varepsilon*PAR*fPAR-Re)*\varphi\_l (B\_l) - k\_l*B\_l$$

where LUE is the Ligth Use Efficiency, $\varepsilon$ takes into account the reduction in LUE due to stress sources, Re is the respiration, ÏȚl(Bl) is the fractional leaf allocation and kl is the leaf natural decay factor to reproduce the senescence. Once Bl is computed it can be transformed into LAI by using the specific leaf area (SLA) and the vegetation fractional cover (fc) according to the next equation:

$$LAI = B\_l*SLA*f\_c$$

In the current version of the manuscript only the references about this model were mentioned and they should be specified. In this way, readers will only have to check the references if they are interested in specific details. These two equations together with the explanation will be provided in the next version of the manuscript. More detailed description can be found in Pasquato et al. (2015) and Ruiz-Pérez et al. (2016) (references embedded in the manuscript).

3- Maybe I missed, but what is the resolution of the implemented model?

You did not miss, we forgot to give that information. The temporal resolution is already specified and it is daily while the spatial resolution was 90X90 meters. It will be included in following versions of the manuscript.

4- How did the manual calibration help to find the best parameters? How the parameters' ranges have been constrained? In table 1, LUE tree and shrub is out of specified range (Shrub is misspelled).

In this case, the manual calibration was considered mandatory as long as the model had never been used at catchment scale and, therefore, we had not clues about its suitability. Although non-statistical indicators were reported, the manual calibration helped to find the best parameters and constrain the searching boundaries in this following three senses: The best set of parameters obtained after the manual calibration was used as seed for the automatic calibration. We think this fact reduced the computational time required by the automatic calibration as long as this starting point or seed is supposed to be closer to the best global solution than a random starting point. We were allowed to double-ckeck the values of the parameters after the manual calibration with those ones recommended in literature. In this way, we assured that the searching boundaries to be used during the automatic calibration process were consistent and wide enough. The manual calibration pointed out that wider ranges were not required and, in this sense, it constrained this searching boundaries. A manual calibration always gives clues about the potential inter-relationships between parameters. These

clues can be used to guide the automatic calibration process (this research was not the case) and to be critic with the results obtained after the automatic calibration (it was the case here) since a sense of relative values was provided by the manual calibration. In that sense, the manual calibration can be extremely helpful to find the best and with physical consistency parameters. Finally, thanks for the observation about Table three. The boundary for all three cases was 1.12 instead of 1.2 and 'Srhub' will be corrected in the whole table.

5- A clearer explanation regarding EOFi would be appreciated. What does different i exactly mean?

If we apply the EOF decomposition (also called Principal Component Analysis) to a simple matrix, the EOFi is the i eigenvector. We always assume that the eigenvectors are ordered according to their corresponding eigenvalues (i.e. the amount of variance explained by them). Hence, EOF1 is the first eigenvector associated with the first eigenvalue and, therefore, which explained more amount of variance. Therefore, i means the position of the eigenvalue when is sorted according the explained variance. In our research, however, we wanted to apply this methodology to analyse spatio-temporal data. That's why the first step was to transform this data into a matrix. Basically, we construct a matrix (F) in which each column is the temporal variation of the data in a particular cell while each row represents the cells values during a particular time step. Once the matrix was constructed, we applied then the EOF analysis as usual. Therefore, we obtained the eigenvectors as usual. However, these eigenvectors can be regarded as maps by considering the same ordering criterion as used in F construction. In this way, the i-eigenvector becomes to the i-main/principal pattern/map. Hence, EOFi is the principal pattern associated with the i eigenvalue. Having reviewed the current manuscript, we found inconsistencies in line 5 and equation 5. We should have kept the same sub-index i instead of j. Otherwise, it might be confusing. We will improve this section and we will check the mathematical consistency within the equations.

6- How would be the model performance with and without calibration on observed satellite data? Any gains or losses there? This would be great to be addressed.

We completely agree with your suggestions. In fact, we are working on it in new on-going projects. In this new applications, we want to use different sources of information (field observations, remote sensing data, etc.) with different resolutions (point measurements, spatio-temporal data, etc.) in order to determine whether models performance improve. However, the study area of this manuscript was discarded for this analysis because this Kenyan catchment can be considered as scarce-data catchment. In fact, the available data is really poor and for this reason, it was precisely selected for this experience. We wanted to face the issue of no having available observations. The calibration was completely 'blind' in terms of observed discharge, i.e. observed discharge was not even known at the beginning of this research. In this way, we assured that the calibration relied only on the satellite data. The main reason to do so was because we did not want to analyse the potential performance improvement by including satellite data, but how far we can arrive by using ONLY satellite data when this data is used as properly as possible. This latter goal leaded and defined all the strategy followed in this research. Anyway, as mentioned, we also are interested in your suggestion but we would recommend to achieve this goal in study areas with good quality of field data. Hopefully, we can discuss in-depth this topic in following applications.

Note: A suplementary file is attached with all this content and proper equations

Please also note the supplement to this comment:
http://www.hydrol-earth-syst-sci-discuss.net/hess-2016-573/hess-2016-573-SC1-supplement.pdf

---

## Referee Comment (RC2) · Anonymous Referee #2 · 16 Feb 2017

Strength -The paper has good scientific argument in validating model outputs in data scarce regions.

Weakness: 1. This draft paper has major language problem. It is recommended that the paper should be edited by professional language editor before the last edition. 2. Referencing problem: the author should be aware scientific referencing. Example: Page 5, section 3: Paragraph two: "Soil texture ranges from sandy clay to clay soils (according to the 1980 UNESCO Soil Map)". 3. What does it mean "Daily precipitation and temperature from 1959 to 2003 were validated by Franz et al. (2010)."??? Page 5,

section 3: Paragraph three.

---

## Short Comment (SC2) · 20 Feb 2017

It is stimulating and motivating to read that the manuscript strength is its scientific argument. As first author, I played the role of main writer and I am obviously not native English speaker. I tried my hardest to mimic English constructions, to be precise with the vocabulary and to reach international English standards. However, I am aware there may be mistakes and it could be a pity to not be precise or understandable due to language issues. As you suggested, the manuscript will be improved either by a professional language editor or by taking advantage of one of the co-authors on board

who is native English speaker.

Regarding to the scientific referencing suggestions:

Page 5, section 3: Paragraph two: "Soil texture ranges from sandy clay to clay soils (According to the 1980 UNESCO Soil Map)" –> 'according to the' will be removed or changed by 'Data source: 1980 UNESCO Soil Map'

Page 5, section 3: Paragraph three. "Daily precipitation and temperature from 1959 to 2003 were validated by Franz et al. (2010)."–> Completely agreed. The term 'validated' is ambiguous. It will be changed by 'statistically analysed'

---

## Referee Comment (RC3) · S. Gharari (Referee) · 17 Mar 2017

I would like to thank the authors for their response to my earlier comments/questions. Please change the manuscript so that it reflects the clarifications regarding to my previous comments/questions.

I would like to add couple of suggestions and questions as reading the manuscript again:

1- I highly recommend the authors to make sure that the sentences are accurate, quantitative and fluent. As an example, in the abstract I can see that the authors wrote "ex-

traordinary amount of information". What does it mean? They also mentioned "scarce data dry region"; do they mean data-scarce dry regions? For example on page 4 line 2 the authors stated that "but it was complete enough for our purpose". What is complete enough and what is the purpose? Is it really necessary to write this sentence? There are many similar cases across the manuscript.

2- I encourage the authors to show the added value of the manuscript clearly and in precise manner. At this moment the manuscript is a mix of methods, literature review and theories. The clarification on model structure, model inputs, model outputs, and the ranges of the parameters would be highly appreciated.

3- I am not convinced that what the authors are showing is only taking into account the remote sensing data. Did the authors look into the seasonality or the recession of the hydrograph and adjust the range accordingly based on some expert guess? If yes, what is the effect of those assumptions or limitations? In a nutshell I would like to see "how exclusive the model result is regarding NDVI".

I believe the manuscript is valuable however major revision seems inevitable.

With regards

Shervan Gharari

---

## Editor Comment (EC1) · D. Solomatine (Editor) · 31 Mar 2017

We can see here an example of an active, enthusiastic discussion, with sincere interest of all parties. Using new data sources (heterogeneous, non-point, with varying uncertainty) is currently an intersting research area with a lot of potential for practical usage - so this paper is a very welcome addition. Authors are invited to revise the manuscript, reply to comments (which was already mainly done during discussion) and submit the manuscript for the re-review.
* * *

---

## Author Comment (AC1) · 31 Mar 2017

First of all, we would want to thank the referees for the suggestions and comments. They will be useful to improve the manuscript. To better address their concerns, our reply has been divided into two sections, concerning: (1) research comments and (2) writing style comments. We are aware that some changes are required. However, both referees highlighted the good research quality of the manuscript and they found it relevant. It is always a pleasure to hear that your work is receiving attention and interest and is providing new insights. The revised version of the manuscript will be

significantly improved incorporating reviewer's suggestions. In the following, we have addressed each question formulated by the referees.

Research comments

1- Are the model parameters different from cell to cell? If yes, which parameters are identical and which parameters are different?

To answer this question is important to better understand the concept of split-structure for the effective parameter value at each cell. This calibration strategy consists on an application of a scalar multiplier to each prior parameter field (specified from data describing watershed characteristics: soils, vegetation, topography, land use, etc.) and to estimate a "best" value for this multiplier via calibration. This so-called "multiplier" approach makes the assumption that the prior parameter field properly describes the spatial pattern of a specific parameter (the pattern of relative magnitudes from cell to cell), but that the magnitudes of all the parameter values must be adjusted to achieve a better simulation of the model response. Hence, the effective parameter at each cell (i.e. the parameter value used when running the model) is compounded by two parts: (1) a common correction factor for each type of parameter that takes into account the model, information and input errors and the temporal and spatial scale effects; and (2) the a priori estimated parameter value at each cell. Hence, for a given parameter, the a priori and effective values are different from cell to cell while the correction factor is common for all cells (and different from map to map). The estimated parameter values were extracted from the field work done and presented in the doctoral thesis by Franz (2007) and following the recommendations provided by the TETIS model's support team. Two of the authors are actually active members of this team and we also used our own experience.

2- I did not understand how the model calculate the LAI which then is used to calculate the transpiration?

The LAI is calculated by the dynamic vegetation sub-model called LUE-Model. The

LUE-Model computes the leaf biomass (Bl) according to the following equation:
$(dB\_l)/dt=(LUE*\varepsilon*PAR*fPAR-Re)*\varphi\_l\ (B\_l)-k\_l*B\_l$

where LUE is the Ligth Use Efficiency, $\varepsilon$ takes into account the reduction in LUE due to stress sources, Re is the respiration, ÏŢl(Bl) is the fractional leaf allocation and kl is the leaf natural decay factor to reproduce the senescence. Once Bl is computed it can be transformed into LAI by using the specific leaf area (SLA) and the vegetation fractional cover (fc) according to the next equation: $LAI=B\_l*SLA*f\_c$ In the current version of the manuscript only the references about this model were mentioned and they should be specified. In this way, readers will only have to check the references if they are interested in specific details. These two equations together with the explanation will be provided in the next version of the manuscript. More detailed description can be found in Pasquato et al. (2015) and Ruiz-Pérez et al. (2016) (references embedded in the manuscript).

3- Maybe I missed, but what is the resolution of the implemented model?

You did not miss, we forgot to give that information. The temporal resolution is already specified and it is daily while the spatial resolution was 90X90 meters. It will be included in following versions of the manuscript.

4- How did the manual calibration help to find the best parameters? How the parameters' ranges have been constrained? In table 1, LUE tree and shrub is out of specified range (Shrub is misspelled).

In this case, the manual calibration was considered mandatory as long as the model had never been used at catchment scale and, therefore, we had not clues about its suitability. Although non-statistical indicators were reported, the manual calibration helped to find the best parameters and constrain the searching boundaries in this following three senses: The best set of parameters obtained after the manual calibration was used as seed for the automatic calibration. We think this fact reduced the computational time required by the automatic calibration as long as this starting point or seed

is supposed to be closer to the best global solution than a random starting point. We were allowed to double-ckeck the values of the parameters after the manual calibration with those ones recommended in literature. In this way, we assured that the searching boundaries to be used during the automatic calibration process were consistent and wide enough. The manual calibration pointed out that wider ranges were not required and, in this sense, it constrained this searching boundaries. A manual calibration always gives clues about the potential inter-relationships between parameters. These clues can be used to guide the automatic calibration process (this research was not the case) and to be critic with the results obtained after the automatic calibration (it was the case here) since a sense of relative values was provided by the manual calibration. In that sense, the manual calibration can be extremely helpful to find the best and with physical consistency parameters. Finally, thanks for the observation about Table three. The boundary for all three cases was 1.12 instead of 1.2 and 'Srhub' will be corrected in the whole table.

5- A clearer explanation regarding EOFi would be appreciated. What does different i exactly mean?

If we apply the EOF decomposition (also called Principal Component Analysis) to a simple matrix, the EOFi is the i eigenvector. We always assume that the eigenvectors are ordered according to their corresponding eigenvalues (i.e. the amount of variance explained by them). Hence, EOF1 is the first eigenvector associated with the first eigenvalue and, therefore, which explained more amount of variance. Therefore, i means the position of the eigenvalue when is sorted according the explained variance. In our research, however, we wanted to apply this methodology to analyse spatio-temporal data. That's why the first step was to transform this data into a matrix. Basically, we construct a matrix (F) in which each column is the temporal variation of the data in a particular cell while each row represents the cells values during a particular time step. Once the matrix was constructed, we applied then the EOF analysis as usual. Therefore, we obtained the eigenvectors as usual. However, these eigenvectors

can be regarded as maps by considering the same ordering criterion as used in F construction. In this way, the i-eigenvector becomes to the i-main/principal pattern/map. Hence, EOFi is the principal pattern associated with the i eigenvalue. Having reviewed the current manuscript, we found inconsistencies in line 5 and equation 5. We should have kept the same sub-index i instead of j. Otherwise, it might be confusing. We will improve this section and we will check the mathematical consistency within the equations. This concepts will be clarified in the revised version of the manuscript.

6- How would be the model performance with and without calibration on observed satellite data? Any gains or losses there? This would be great to be addressed.

We completely agree with your suggestions. In fact, we are working on it in new ongoing projects. In this new applications, we want to use different sources of information (field observations, remote sensing data, etc.) with different resolutions (point measurements, spatio-temporal data, etc.) in order to determine whether models performance improve. However, the study area of this manuscript was discarded for this analysis because this Kenyan catchment can be considered as scarce-data catchment. In fact, the available data is really poor and for this reason, it was precisely selected for this experiment. We wanted to face the issue of no having available observations. The calibration was completely 'blind' in terms of observed discharge, i.e. observed discharge was not even known at the beginning of this research. In this way, we assured that the calibration relied only on the satellite data. The main reason to do so was because we did not want to analyse the potential performance improvement by including satellite data, but how well we can calibrate a model by using only satellite data when this data is used properly. This latter goal builds the main theme of this research. Anyway, as mentioned, we also are interested in your suggestion but we would recommend to achieve this goal in study areas with good quality of field data. Hopefully, we can discuss in-depth this topic in following applications.

7- I am not convinced that what the authors are showing is only taking into account the remote sensing data. Did the authors look into the seasonality or the recession of

the hydrograph and adjust the range accordingly based on some expert guess? If yes, what is the effect of those assumptions or limitations? In a nutshell I would like to see "how exclusive the model result is regarding NDVI".

As said in the previous question, the calibration of the model was blind in terms of observed discharge. We did not use the observed discharge in any way. Neither the observed discharge per se nor its seasonality and/or other statistical metric. Therefore, we did not look into the seasonality of the recession of the hydrograph, neither adjust the range accordingly based on some expert guess. However, the TETIS-VEG model is process based. It is not a black box and it was driven by precipitation and temperature records measured at field. The estimated value at each cell was done by using data describing watershed characteristics (soils, vegetation, topography, land use, etc.). Moreover, the proposed calibration process relied on the satellite NDVI main patterns. As said in page 15 lines 1 to 3, the temporal variation of the EOF1 (which explained more than 60% and which dominated the calibration process) was related to the two usual rainy seasons of the study area. The NDVI contains information about seasonality by itself. By using the proposed conceptual model, such information is transferred to all hydrological processes involved in the water cycle (incl. discharge at the outlet point). Of course, some characteristics as runoff propagation parameters cannot be assessed using satellite NDVI. However, they are not influent in this case study since the model was run at daily time step.

Writing style comments

Since these comments are very similar, we consider more fruitful to address them with a common response. These are the comments regarding to language issues and style: This draft paper has major language problem. It is recommended that the paper should be edited by professional language editor before the last edition. I highly recommend the authors to make sure that the sentences are accurate, quantitative and fluent. As an example, in the abstract I can see that the authors wrote "extraordinary amount of information". What does it mean? They also mentioned "scarce data dry region"; do

they mean data-scarce dry regions? For example on page 4 line 2 the authors stated that "but it was complete enough for our purpose". What is complete enough and what is the purpose? Is it really necessary to write this sentence? There are many similar cases across the manuscript. I encourage the authors to show the added value of the manuscript clearly and in precise manner. At this moment the manuscript is a mix of methods, literature review and theories. The clarification on model structure, model inputs, model outputs, and the ranges of the parameters would be highly appreciated. As suggested by the reviewers, the manuscript will be improved either by a professional language editor or by taking advantage of one of the co-authors on board who is native English speaker Additionally, we will give a thorough editorial check in order to meet the requested requirements. We will avoid 'empty' sentences as those ones mentioned in the second bullet. We will ensure that the sentences are accurate, quantitative and fluent. To accomplish the last bullet, we will improve some sections and some changes in the document organization are likely to happen.

Note: A supplementary file is attached with all this content and proper equations

Please also note the supplement to this comment:
http://www.hydrol-earth-syst-sci-discuss.net/hess-2016-573/hess-2016-573-AC1-supplement.pdf

---

## Author Response (AR1)

**POINT-BY-POINT RESPONSE**

First of all, we would want to thank the referees for the suggestions and comments. They were highly useful to improve the manuscript. To better address their concerns, their comment were divided into two sections, concerning: (1) research comments and (2) writing style comments.

Since both referees highlighted the good research quality of the manuscript and they found it relevant, we think the revised version of the manuscript provides new insights and innovative calibration tools and HESS would be a perfect journal to disseminate the outcomes of this research.

In the following, we address each question formulated by the referees.

**Research comments**

*1- Are the model parameters different from cell to cell? If yes, which parameters are identical and which parameters are different?*

To answer this question is important to better understand the concept of split-structure for the effective parameter value at each cell. This calibration strategy consists on an application of a scalar multiplier to each prior parameter field (specified from data describing watershed characteristics: soils, vegetation, topography, land use, etc.) and to estimate a "best" value for this multiplier via calibration. This so-called "multiplier" approach makes the assumption that the prior parameter field properly describes the spatial pattern of a specific parameter (the pattern of relative magnitudes from cell to cell), but that the magnitudes of all the parameter values must be adjusted to achieve a better simulation of the model response.

Hence, the effective parameter at each cell (i.e. the parameter value used when running the model) is compounded by two parts: (1) a common correction factor for each type of parameter that takes into account the model, information and input errors and the temporal and spatial scale effects; and (2) the *a priori* estimated parameter value at each cell.

Therefore, for a given parameter, the *a priori* and effective values are different from cell to cell while the correction factor is common for all cells (and different from map to map). The estimated parameter values were extracted from the field work done and presented in the doctoral thesis by Franz (2007) and following the recommendations provided by the TETIS model's support team. Two of the authors are actually active members of this team and we also used our own experience.

This discussion is included in the current version of the manuscript from line 13 to line 19, page 7.

*2- I did not understand how the model calculate the LAI which then is used to calculate the transpiration?*

The LAI is calculated by the dynamic vegetation sub-model called LUE-Model. The LUE-Model computes the leaf biomass ($B_l$) according to the following equation:

$$\frac{dB_l}{dt} = (LUE * \varepsilon * PAR * fPAR - Re) * \varphi_l(B_l) - k_l * B_l$$

where LUE is the Ligth Use Efficiency, $\varepsilon$ takes into account the reduction in LUE due to stress sources, Re is the respiration, $\phi_l(B_l)$ is the fractional leaf allocation and $k_l$ is the leaf natural decay factor to reproduce the senescence.

Once $B_l$ is computed it can be transformed into LAI by using the specific leaf area (SLA) and the vegetation fractional cover ($f_c$) according to the next equation:

$$LAI = B_l * SLA * f_c$$

In the last version of the manuscript only the references about this model were mentioned. This more detailed explanation is provided in the current version of the manuscript (equation 1 and text from line 30 to 32, page 7). In this way, readers will only have to check the references if they are interested in specific details. More detailed description can be found in Pasquato *et al.* (2015) and Ruiz-Pérez *et al.* (2016) (references embedded in the manuscript).

*3- Maybe I missed, but what is the resolution of the implemented model?*
You did not miss, we forgot to give that information. The temporal resolution is already specified and it is daily while the spatial resolution was 90X90 meters. It is included in the current manuscript in line 32, page 6.

*4- How did the manual calibration help to find the best parameters? How the parameters' ranges have been constrained? In table 1, LUE tree and shrub is out of specified range (Shrub is misspelled).*
In this case, the manual calibration was considered mandatory as long as the model had never been used at catchment scale and, therefore, we had not clues about its suitability. Although non-statistical indicators were reported, the manual calibration helped to find the best parameters and constrain the searching boundaries in this following three senses:

1. The best set of parameters obtained after the manual calibration was used as seed for the automatic calibration. We think this fact reduced the computational time required by the automatic calibration as long as this starting point or seed is supposed to be closer to the best global solution than a random starting point.
2. We were allowed to double-ckeck the values of the parameters after the manual calibration with those ones recommended in literature. In this way, we assured that the searching boundaries to be used during the automatic calibration process were consistent and wide enough. The manual calibration pointed out that wider ranges were not required and, in this sense, it constrained this searching boundaries.
3. A manual calibration always gives clues about the potential inter-relationships between parameters. These clues can be used to guide the automatic calibration process (this research was not the case) and to be critic with the results obtained after the automatic calibration (it was the case here) since a sense of relative values was provided by the manual calibration. In that sense, the manual calibration can be extremely helpful to find the best and with physical consistency parameters.

Finally, thanks for the observation about Table three. The boundary for all three cases was 1.12 instead of 1.2 and 'Srhub' was corrected in the whole table.

*5- A clearer explanation regarding EOF$_i$ would be appreciated. What does different i exactly mean?*
If we apply the EOF decomposition (also called Principal Component Analysis) to a simple matrix, the EOF$_i$ is the i eigenvector. We always assume that the eigenvectors are ordered according to their corresponding eigenvalues (i.e. the amount of variance explained by them). Hence, EOF$_1$ is the first eigenvector associated with the first eigenvalue and, therefore, which explained more amount of variance. Therefore, i means the position of the eigenvalue when is sorted according the explained variance.

In our research, however, we wanted to apply this methodology to analyse spatio-temporal data. That's why the first step was to transform this data into a matrix. Basically, we construct a matrix (F) in which each column is the temporal variation of the data in a particular cell while each row represents the cells values during a particular time step. Once the matrix was constructed, we applied then the EOF analysis as usual. Therefore, we obtained the eigenvectors as usual. However, these eigenvectors can be regarded as maps by considering the same ordering criterion as used in F construction. In this way, the i-eigenvector becomes to the i-main/principal pattern/map. Hence, $EOF_i$ is the principal pattern associated with the i eigenvalue.

Having reviewed the current manuscript, we found inconsistencies in line 5 and equation 5. We should have kept the same sub-index i instead of j. Otherwise, it might be confusing. We checked the mathematical consistency within the equations as can be seen in equation 5 and lines 31 and 32, page 9.

*6- How would be the model performance with and without calibration on observed satellite data? Any gains or losses there? This would be great to be addressed.*
We completely agree with your suggestions. In fact, we are working on it in new on-going projects. In this new applications, we want to use different sources of information (field observations, remote sensing data, etc.) with different resolutions (point measurements, spatio-temporal data, etc.) in order to determine whether models performance improve. However, the study area of this manuscript was discarded for this analysis because this Kenyan catchment can be considered as scarce-data catchment. In fact, the available data is really poor and for this reason, it was precisely selected for this experiment. We wanted to face the issue of no having available observations. The calibration was completely 'blind' in terms of observed discharge, i.e. observed discharge was not even known at the beginning of this research. In this way, we assured that the calibration relied only on the satellite data. The main reason to do so was because we did not want to analyse the potential performance improvement by including satellite data, but how well we can calibrate a model by using only satellite data when this data is used properly. This latter goal builds the main theme of this research.

Anyway, as mentioned, we also are interested in your suggestion but we would recommend to achieve this goal in study areas with good quality of field data. Hopefully, we can discuss in-depth this topic in following applications.

*7- I am not convinced that what the authors are showing is only taking into account the remote sensing data. Did the authors look into the seasonality or the recession of the hydrograph and adjust the range accordingly based on some expert guess? If yes, what is the effect of those assumptions or limitations? In a nutshell I would like to see "how exclusive the model result is regarding NDVI".*
As said in the previous question, the calibration of the model was blind in terms of observed discharge. We did not use the observed discharge in any way. Neither the observed discharge *per se* nor its seasonality and/or other statistical metric. Therefore, we did not look into the seasonality of the recession of the hydrograph, neither adjust the range accordingly based on some expert guess. However, the TETIS-VEG model is process based. It is not a black box and it was driven by precipitation and temperature records measured at field. The estimated value at each cell was done by using data describing watershed characteristics (soils, vegetation, topography, land use, etc.). Moreover, the proposed calibration process relied on the satellite NDVI main patterns. As said in page 15 lines 1 to 3, the temporal variation of the $EOF_1$ (which explained more than 60% and which dominated the calibration process) was related to the two usual rainy seasons of the study area. The NDVI contains information about seasonality by itself. By using the proposed conceptual model, such information is transferred

to all hydrological processes involved in the water cycle (incl. discharge at the outlet point). Of course, some characteristics as runoff propagation parameters cannot be assessed using satellite NDVI. However, they are not influent in this case study since the model was run at daily time step.

**Writing style comments**

Since these comments are very similar, we consider more fruitful to address them with a common response. These are the comments regarding to language issues and style:

- This draft paper has major language problem. It is recommended that the paper should be edited by professional language editor before the last edition.
- I highly recommend the authors to make sure that the sentences are accurate, quantitative and fluent. As an example, in the abstract I can see that the authors wrote "extraordinary amount of information". What does it mean? They also mentioned "scarce data dry region"; do they mean data-scarce dry regions? For example on page 4 line 2 the authors stated that "but it was complete enough for our purpose". What is complete enough and what is the purpose? Is it really necessary to write this sentence? There are many similar cases across the manuscript.
- I encourage the authors to show the added value of the manuscript clearly and in precise manner. At this moment the manuscript is a mix of methods, literature review and theories. The clarification on model structure, model inputs, model outputs, and the ranges of the parameters would be highly appreciated.

The manuscript was improved by taking advantage of one of the co-authors on board who is native English speaker.

Additionally, we gave a thorough editorial check in order to meet the requested requirements. We removed 'empty' sentences as those ones mentioned in the second bullet. We ensured that the sentences are accurate, quantitative and fluent. All these improvements can be seen along the marked-up manuscript version.

**LIST OF RELEVANT CHANGES**

- In-depth explanation of the split-structure for the effective parameter used in the TETIS-VEG model.
- Improvement of the mathematical equations of the EOF methodology.
- More detailed description of how LAI is simulated in the proposed dynamic vegetation model.
- Thorough editorial check along the manuscript in order to meet the requirements requested by the reviewers.
- Use of more direct and fluent sentences. In particular, the conclusion section was substantially modified and improved.
- Minor errors (such as language-related errors, typos, etc) were also changed

[revised manuscript text omitted]

---

## Referee Report (RR1)

Did the authors try to use the LAI product instead of NDVI? They state the reason for using NDVI which I do not dispute but I was just curious to know if they have done some testing with LAI or if they plan to do so in the future.

Minor comments:

1) Line 10 page 2: the authors state that many studies observed that Eta is a major diver of hydrological records but only cite one publication.
2) I would shorten the first paragraph of section 2 (lines 22-29, page 3). I believe the authors can assume that the reader knows how Web of Knowledge works.
3) Line 7, page 4: more informative than "(the US-German satellite mission)" could be a short description of which data GRACE could provide for hydrological models.
4) Line 14, page 4: I suggest to remove "that"
5) Line 23, page 4: Please consider checking and reformulating the sentence "Those points were selected randomly or by considering the knowledge about each study site".
6) Line 28, page 4: I suggest to replace "at" with "of".
7) Line 6, page 5: I suggest to add "surfaces" between "remaining" and "of", and to replace "is" with "are"
8) Line 14, page 5: what do the authors mean with "phase of the year"?
9) Line 17-18, page 5: I would remove the sentence that mentions the tiles used in the study.
10) Lines 19-20, page 5: I would rephrase the sentence as follow: "The used NDVI products (MOD13Q1 and MYD13Q1) are level 3 products that means they are note raw satellite data."
11) Line 20, page 5: I would remove "Actually"
12) Line 25, page 5: I would replace "experience" with "experiences" and add a space after the bracket
13) Line 30, page 5: I would avoid contractions. You could replace "that's why" with "we therefore decided…"
14) General remark on the description of the used NDVI product from MODIS: I found the text a bit too long and too technical. NDVI product from MODIS is a widely used RS product and the authors do not need to prove its scientific base.
15) Line 19, page 7: I believe that not all the terms of the equation are described in the text.
16) Line 29, page 7: what are "modelling evaluations"?
17) Lines 24-25, page 9: the sentence is not clear.
18) Line 13, page 10: I would add "but" between "content" and "referred".
19) Lines 10-12, page 14: The fit between simulated and observed flow seems to improve with time. Could this fact be also related with the amount of flow (there are sensibly higher peaks 2002 than in 2000)?
20) Line 21, page 16: the use of the word "enormous" seems a bit extreme to me.
21) Line 26, page 16: the expression "in order" is used twice in the same sentence
22) Figure 1: coordinates and scale are missing. I would reduce the size of the African continent and increase the size of the analysed basin and perhaps add some more spatial information instead of just the boundaries. For instance, the author could use the elevation map (which is governing the rainfall patterns) and overlay the basin boundary and the rainfall stations.

---

## Author Response (AR2)

**#REFEREE 1**

**The manuscript is improved significantly however my only remaining concerns is regarding the parameter sets of the model and their limits. I did not fully understand how the ranges were chosen and if prior knowledge was used in choosing those boundaries for parameter sets. If prior knowledge is used in selecting the ranges then the statement that the model is fully calibrated on remote sensing data is not entirely correct. In better words I would like to see the model simulation given the maximum parameter ranges possible and with no manual calibration (no prior knowledge included) compared with the case presented in the manuscript.**

Firstly, we would thank the involvement of referee #1 during the revision process. The discussion with this referee has been fruitful and the current form of the paper has improved substantially thanks to his comments (see public discussion for further details). His only remaining concern is regarding the role played by the manual calibration and whether it could have been masking the obtained results. He wonders whether we used prior information to performance the manual calibration because, if so, the statement claiming that we only used remote sensing data is not correct.

As discussed in the first round of comments and specified in the sections 5.2 and 5.3, the automatic and the manual calibration were blind, i.e. we did not use any variable measured at field, neither prior information for the correction factors to be calibrated. Moreover, the range of variation of the factors to be calibrated was broad. Indeed, we employed the maximum range of variation suggested by the TETIS model developers.

Both calibration approaches, manual and automatic, maximized the correlation between simulated LAI and satellite NDVI. None of them involved neither field measurements nor prior information of the correction factors to calibrate. The answer is thus clear: we did not use prior information about the correction factors included in the manual/automatic calibration process and both calibration processes only relied on satellite data, particularly on NDVI from MODIS. The main difference between the two approaches is that while the manual calibration consisted on the usual ad hoc method (manual adjustment of parameter values) in some selected cells, the proposed automatic calibration considered the whole catchment and was able to take advantage of all spatio-temporal information contained in the satellite NDVI data.

Moreover, the results obtained by the manual calibration were only used as seed to start the automatic calibration. We did not use any additional information from the manual calibration. We kept the same range of variation of the parameters. This manual calibration was actually meant to check that the model was able to obtain acceptable results without showing any bias. It was highly useful since the model (including the vegetation module) had never been used at

catchment scale. According to Pham and Karaboga (2012), manual calibration and expert supervision of any automatic calibration must be always applied.

As said in the manuscript, we used a genetic algorithm (GA) to perform the automatic calibration – Pyevolve in particular. At the start of optimization, a GA requires a group of initial solutions. There are two ways of forming this initial population. The first consists of using randomly produced solutions created by a random number generator. The second one method employs solutions that already provide acceptable results to form an initial population. According to Pham and Karaboga (2012), the only difference between these two approaches is that the GA converges to an optimal solution in less time by applying the second approach. But, both approaches will provide the same result as long as the optimization algorithm provides an actual global optimum solution. GA has shown great ability to provide global optimal solutions, dodging local ones. And, in particular, PyEvolve has been successfully applied broadly (e.g. Butterfield et al., 2004). We inclined to use the second approach and we used the results obtained by the manual calibration as initial population for the automatic calibration, being this the only information transferred from the manual to the automatic calibration.

We consider thus this manual calibration as good practice because it might reduce the computational time consumed for the automatic calibration (factor to consider when working with spatio-temporal data). It provides information about results of the model in points within the study site in addition to the general result at catchment scale provided by the automatic calibration. It relied only on satellite information. In overall, we believe that this manual calibration does not compromise the statement claiming that this model is only calibrated based on remote sensing data (because it was actually calibrated only using remote sensing), which is the main objective of this research.

**_REFERENCES_**

Pham, D., & Karaboga, D. (2012). _Intelligent optimisation techniques: genetic algorithms, tabu search, simulated annealing and neural networks._ Springer Science & Business Media.

Butterfield, A., Vedagiri, V., Lang, E., Lawrence, C., Wakefield, M. J., Isaev, A., & Huttley, G. A. (2004). PyEvolve: a toolkit for statistical modelling of molecular evolution. _BMC bioinformatics_, 5(1), 1.

**#REFEREE 2**

**Did the authors try to use the LAI product instead of NDVI? They state the reason for using NDVI which I do not dispute but I was just curious to know if they have done some testing with LAI or if they plan to do so in the future.**

First of all, we would like to thank the anonymous referee for his/her review. We are very pleased to have received an overall positive evaluation of our manuscript and will gladly revise it following his/her comments to further strengthen the scientific quality of our work. We hope that the changes in the revised manuscript will be well received.

In reference to your question, we did not try to use the LAI product in this specific application. The main author did try to use LAI in other study case with no successful results (see details in Ruiz-Pérez *et al.,* 2016). Moreover, other authors (e.g. Pasquato *et al.,* 2015) found discrepancies between field and satellite based LAI. MODIS LAI products are generated by a model and this model requires spatial information such as land use coverage. These products are therefore subject to the inherent uncertainty due to model conceptual limitation plus the uncertainty attached to the required spatial information. We therefore felt more confident by using 'pure' satellite data, i.e. data not generated by model.

We do not plan to use this specific product in the future. However, all we are working on remote sensing data and its applicability for vegetation monitoring. We are certain that new insights at this regard will come as result of our currents projects.

**Minor comments:**

**1) Line 10 page 2: the authors state that many studies observed that Eta is a major driver of hydrological records but only cite one publication.**

More references have been included.

**2) I would shorten the first paragraph of section 2 (lines 22-29, page 3). I believe the authors can assume that the reader knows how Web of Knowledge works.**

The paragraph has been shortened as suggested by the referee. We have only kept the basic information.

**3) Line 7, page 4: more informative than "(the US-German satellite mission)" could be a short description of which data GRACE could provide for hydrological models.**

We have added some information and an additional reference.

**4) Line 14, page 4: I suggest to remove "that"**

It has been removed.

**5) Line 23, page 4: Please consider checking and reformulating the sentence "Those points were selected randomly or by considering the knowledge about each study site".**

The sentence has been removed. It was confusing and did not provide any important detail.

**6) Line 28, page 4: I suggest to replace "at" with "of".**

It has been replaced

**7) Line 6, page 5: I suggest to add "surfaces" between "remaining" and "of", and to replace "is" with "are"**

All changes have been done

**8) Line 14, page 5: what do the authors mean with "phase of the year"?**

This term has been removed. As pointed by the referee, it was ambiguous.

**9) Line 17-18, page 5: I would remove the sentence that mentions the tiles used in the study.**

The sentence has been removed

**10) Lines 19-20, page 5: I would rephrase the sentence as follow: "The used NDVI products (MOD13Q1 and MYD13Q1) are level 3 products that means they are note raw satellite data."**

The sentence has been changed as suggested

**11) Line 20, page 5: I would remove "Actually"**

It has been removed

**12) Line 25, page 5: I would replace "experience" with "experiences" and add a space after the bracket**

It has been done

**13) Line 30, page 5: I would avoid contractions. You could replace "that's why" with "we therefore decided…"**

We have checked the document and replaced contractions with either the full form or alternative expressions

**14) General remark on the description of the used NDVI product from MODIS: I found the text a bit too long and too technical. NDVI product from MODIS is a widely used RS product and the authors do not need to prove its scientific base.**

This part has been reduced as suggested. In particular, we have removed all technical information of low quality pixels removal.

**15) Line 19, page 7: I believe that not all the terms of the equation are described in the text.**

Now, they are all described

**16) Line 29, page 7: what are "modelling evaluations"?**

The sentence has been reformulated

**17) Lines 24-25, page 9: the sentence is not clear.**

It has been reformulated

**18) Line 13, page 10: I would add "but" between "content" and "referred".**

It has been added

**19) Lines 10-12, page 14: The fit between simulated and observed flow seems to improve with time. Could this fact be also related with the amount of flow (there are sensibly higher peaks 2002 than in 2000)?**

The fit between simulated and observed also improves as the amount of flow increases. Based on our experience (Ruiz-Pérez *et al.,* 2016) and supported by other authors (Durand *et al.,* 1992; Parkin *et al.,* 1996; Medici *et al.,* 2008 among others), hydrological models perform better under wet conditions than under dry ones. Indeed, modelling non-perennial rivers is a challenging task as pointed out by Adamowski and Sun (2010). We agree with the referee that the misfit within year 2000 might be also related to the low peak values. However, the comparison of the flow duration curves among the three years showed that the model was able to reproduce high and low discharge values with the same accuracy within the last year. We thus think that the model performance improves with the time as explained in the manuscript.

**20) Line 21, page 16: the use of the word "enormous" seems a bit extreme to me.**

It has been replaced with "high"

**21) Line 26, page 16: the expression "in order" is used twice in the same sentence**

The sentence has been reformulated

**22) Figure 1: coordinates and scale are missing. I would reduce the size of the African continent and increase the size of the analysed basin and perhaps add some more spatial information instead of just the boundaries. For instance, the author could use the elevation map (which is governing the rainfall patterns) and overlay the basin boundary and the rainfall stations.**

Figure 1 has been improved according to the referee's suggestions.

[revised manuscript text omitted]